# SEMANTIC VOTING: A SELF-EVALUATION-FREE APPROACH FOR EFFICIENT LLM SELF-IMPROVEMENT ON UNVERIFIABLE OPEN-ENDED TASKS

**Chunyang Jiang, Yonggang Zhang, Yiyang Cai, Chi-Min Chan, Yulong Liu,**
**Mingming Chen✉, Wei Xue, and Yike Guo✉**
Hong Kong University of Science and Technology
cjaingaq@connect.ust.hk

## ABSTRACT

The rising cost of acquiring supervised data has driven significant interest in self-improvement for large language models (LLMs). Straightforward unsupervised signals like majority voting have proven effective in generating pseudo-labels for verifiable tasks, while their applicability to unverifiable tasks (*e.g.*, translation) is limited by the open-ended character of responses. As a result, self-evaluation mechanisms (*e.g.*, self-judging and entropy minimization) are predominantly used to derive pseudo-labels. However, self-evaluation relying on LLMs typically incurs high computational overhead and introduces overconfidence issues due to intrinsic biases. To address these challenges, we propose a novel self-evaluation-free approach for unverifiable tasks, designed for lightweight yet effective self-improvement. Inspired by majority voting commonly employed in verifiable tasks, we propose semantic voting as a novel mechanism that relaxes the principle of hard matching (*i.e.*, exact matching) toward soft matching (*i.e.*, semantic similarity). Soft matching is achieved by leveraging a lightweight sentence embedding model to quantify semantic similarity, thereby mitigating excessive computational burden and intrinsic bias-associated limitations of self-evaluation. Comprehensive experiments demonstrate that our method achieves substantial gains in computational efficiency and overall better performance than self-evaluation methods across diverse model architectures and tasks. We have released our codes at https://github.com/rubickkcibur/Semantic-Voting.

## 1 INTRODUCTION

Recent advances in large language model (LLM) research and engineering (DeepSeek-AI et al., 2025; Comanici et al., 2025; Yang et al., 2025) have led to a surge in demand for high-cost supervision signals, such as expert-annotated reasoning trajectories (Xia et al., 2025), fine-grained human feedback (Wu et al., 2023), and deliberately designed verifiers (Chervonyi et al., 2025). These supervision methods typically require substantial human effort and often suffer from limited generalizability, resulting in high data acquisition costs (Villalobos et al., 2024). To address these challenges, LLM self-improvement (Huang et al., 2023) has emerged as a promising paradigm, enabling self-training through unsupervised (Zuo et al., 2025) or self-supervised (Yuan et al., 2024) generation of pseudo-labels, thereby breaking free from reliance on external costly supervision.

One of the most prominent self-improvement approaches is building pseudo-labels via majority voting (Wang et al., 2023), which generates multiple response candidates for a given question and selects the most frequently occurring answer as the pseudo-label. This mechanism has proven to be a simple yet effective strategy across diverse training paradigms (Huang et al., 2023; Prasad et al., 2024; Zuo et al., 2025). However, majority voting relies on exact matching, limiting its applicability to closed-ended tasks, such as arithmetic problems and multiple-choice questions.

To realize self-improvement on unverifiable open-ended tasks, two self-evaluation-based strategies are predominantly adopted. The first, self-judging, leverages "LLM-as-a-Judge" paradigm, where

---

✉The corresponding authors are Mingming Chen and Yike Guo.

the model evaluates its own generated responses by assigning scores or preferences, enabling self-rewarding without reliance on external feedback (Yuan et al., 2024). The second is based on entropy minimization, which estimates the confidence of generated outputs using measures such as Shannon entropy and selectively reinforces those with lower entropy (*i.e.*, higher self-confidence) (Prabhudesai et al., 2025; Zhao et al., 2025; Zhang et al., 2025b).

While these self-evaluation approaches aim to construct precise pseudo-labels by leveraging the rich knowledge of large pretrained models, recent studies (Chowdhury et al., 2024) have highlighted the robustness of preference learning in enabling LLMs to improve even from noisy labels. This robustness calls into question the necessity of pursuing fine-grained pseudo-feedback, especially given that self-evaluation methods incur substantial computational overhead due to additional inference steps and introduce overconfidence issues (Li et al., 2024a; Zhang et al., 2025c) that stem from the model's intrinsic biases.

In this paper, we present a lightweight framework for self-improving models on unverifiable open-ended tasks, bypassing the need for self-evaluation mechanisms. Inspired by the principle of majority voting, we introduce *semantic voting*, a novel approach that relaxes the hard matching (*i.e.*, exact matching) to soft matching based on semantic similarities. Specifically, each self-generated response is encoded into a semantic vector using a sentence embedding model, and its voting score is computed as the average cosine similarity with all other responses. Based on these scores, we identify the most and least favored responses to form preference pairs, which serve as pseudo-signals to refine the model through preference learning.

Our experiments on machine translation and text summarization demonstrate that the semantic voting framework consistently enhances performance across diverse model architectures and tasks, outperforming self-evaluation approaches in both stability and efficiency. The computational overhead of semantic voting amounts to only a small fraction (ranging from several thousandths to a few percent) of the time required by self-evaluation baselines. Further analysis confirms that semantic voting provides a meaningful and effective unsupervised signal, going beyond what might otherwise be a "*spurious reward*" (Shao et al., 2025).

## 2 RELATED WORKS

### 2.1 LLM SELF-IMPROVEMENT BY MAJORITY VOTING

Majority voting was initially introduced as a decoding strategy to improve the test-time accuracy (Wang et al., 2023). It decides the final answer by selecting the most frequent one among multiple generated candidates. Inspired by its success, LMSI (Huang et al., 2023) adapted majority voting as a criterion for rejection sampling to enable unsupervised improvement for LLMs. Subsequent studies integrated it into alignment frameworks. ScPO (Prasad et al., 2024) builds preference pairs with the most- and least-voted answers, using Direct Preference Optimization (Rafailov et al., 2023) for model improvement. TTRL (Zuo et al., 2025) shapes rewards via majority voting and optimizes the model with GRPO (Shao et al., 2024). There are also other approaches that leverage majority voting as part of their data-filtering strategies or construction basis for pseudo-labels (Jiang et al., 2025b; Jiao et al., 2025; Zhang et al., 2025a). Despite the widely demonstrated effectiveness of majority voting, its application is strictly limited to tasks where candidate answers can be exactly matched, such as arithmetic or multiple-choice questions.

### 2.2 LLM SELF-IMPROVEMENT BY SELF-JUDGING

Self-judging is rooted in the "LLM-as-a-Judge" paradigm, which leverages LLMs as evaluators to provide cost-effective feedback for complex tasks (Zheng et al., 2023; Gu et al., 2024). In the context of self-improvement, self-judging typically involves instructing the model to assess its own generated responses, often by assigning numerical scores or binary decisions. These self-assessments are then used as pseudo-labels for data filtering or as reward signals in training (Li et al., 2024b; Yuan et al., 2024; Pang et al., 2024; Lee et al., 2024; Zhang et al., 2024b). While self-judging offers a simple and response format-agnostic approach to handle diverse tasks, its impartiality can be undermined by the self-preference bias, which also increases vulnerability to reward hacking (Li et al., 2024a). Moreover, self-judging presumes strong evaluative capabilities, limiting efficacy on smaller

or less capable models (Wang et al., 2025). The generation of detailed evaluation rationales also incurs non-negligible computational overhead, hindering large-scale use (Li et al., 2024a).

### 2.3 LLM SELF-IMPROVEMENT BY ENTROPY MINIMIZATION

Recently, entropy minimization has emerged as a popular self-improvement strategy for LLMs, encouraging models to favor high-confidence (low-entropy) outputs. Agarwal et al. (2025) introduced token- and trajectory-level entropy as two effective RL rewards, demonstrating their efficacy in boosting reasoning performance. Concurrently, RENT (Prabhudesai et al., 2025) confirmed the token-level entropy's utility and found that restricting it to specific segments of outputs further improves performance. INTUITOR (Zhao et al., 2025) instead uses self-certainty, the average KL divergence between the uniform distribution and the model's logits, as a confidence-enhancement reward. GenRM (Li et al., 2025) provides theoretical grounding for entropy minimization: from the Inverse Reinforcement Learning perspective, pretrained logits act as optimal Q-functions, making entropy minimization a natural alignment with pretraining knowledge.

Despite its strong performance on reasoning tasks (*e.g.*, math, coding), entropy minimization exhibits notable inconsistencies across different models and tasks. Agarwal et al. (2025) observed that entropy minimization fails when task bias diverges from pretraining data, misleading confidence toward unhelpful priors. They also reported inconsistency across different models, suggesting a heavy dependence on the base model's inherent capabilities. Zhang et al. (2025c) further highlights the risk of overconfidence, showing that deuced entropy minimization may lead to performance degradation and even model collapse.

## 3 METHODOLOGY

In this section, we introduce our *semantic voting*-based self-improvement (**SVSI**) framework, which assigns pseudo-scores to self-generated responses based on semantic consensus and constructs preference pairs for learning. An overview is shown in Figure 1.

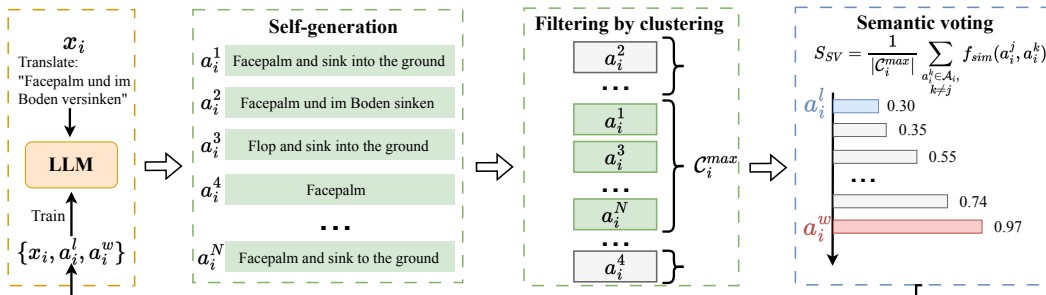

Figure 1: Overview of semantic voting-based self-improvement (SVSI) for LLMs. Given an input question $x_i$, SVSI first generates a candidate answer set $\mathcal{A}_i$, clustering it to identify the most coherent (largest) subset $C_i^{max}$ (Section 3.2), and then performs semantic voting within $C_i^{max}$ using average semantic similarities (Section 3.1). Answers with the highest ($a_i^w$) and lowest ($a_i^l$) voting scores are used for DPO training (Section 3.3).

### 3.1 SEMANTIC VOTING

Given an input question $x_i$ and $N$ stochastic responses sampled from an LLM $\mathcal{M}$, the candidate answer set $\mathcal{A}_i$ is defined as:

$$\mathcal{A}_i = \{a_i^j | a_i^j = a(y_i^j) \sim \mathcal{M}(\cdot|x_i)\}_j^N \tag{1}$$

where $y_i^j$ represents the generated response and $a_i^j = a(y_i^j)$ denotes the predicted answer parsed from $y_i^j$. For simplicity, we use $a_i^j$ throughout.

Similar to majority voting (Wang et al., 2023), semantic voting leverages a consensus mechanism among candidate answers, but instead of hard exact matching, it employs a soft measure via a similarity function $f_{\text{sim}} : \mathcal{A}^2 \rightarrow [0, 1]$:

$$S_{\text{sv}}(a_i^j | \mathcal{A}_i) = \frac{1}{|\mathcal{A}_i|} \sum_{a_i^j, a_i^k \in \mathcal{A}_i, k \neq j} f_{\text{sim}}(a_i^j, a_i^k) \tag{2}$$

The answer with the highest $S_{SV}$ score is thus the one most semantically aligned with the rest of the set, *i.e.*, the "most voted" under semantic consensus.

In this work, we employ a lightweight implementation of the similarity function $f_{sim}$, defined as the cosine similarity between the sentence embeddings of two candidate answers:

$$f_{\text{sim}}(a_i^j, a_i^k) = \frac{\mathcal{M}_{emb}(a_i^j) \cdot \mathcal{M}_{emb}(a_i^k)}{||\mathcal{M}_{emb}(a_i^j)|| \times ||\mathcal{M}_{emb}(a_i^k)||} \tag{3}$$

where $\mathcal{M}_{emb}$ is a sentence embedding model that embeds the answers into semantic vectors.

## 3.2 SELF-GENERATION FILTERING

Intuitively, semantic voting is to identify a semantic consensus among candidate answers and score each candidate by its alignment with such shared consensus. Therefore, to ensure meaningful and reliable consensus formation, the candidate pool should exhibit relatively limited variation; otherwise, severely deviant samples may skew the voting outcome. While the candidate answers are generated in response to the same question and thus expected to be basically analogous, it remains possible for stochastic self-generation to produce outliers that diverge significantly from the majority. Our empirical observation also confirms that such anomalous candidates can undermine the effectiveness of semantic voting, particularly for less capable models.

To address this problem and facilitate the stability of self-improvement, we filter the self-generated answers by a clustering process before semantic voting. Specifically, we apply a density-based clustering algorithm (Malzer & Baum, 2020) to aggregate the candidate answer set $\mathcal{A}_i$ into one or more clusters $\{C_i^k\}_k$, with their sentence embeddings as features and the cosine similarity as distance. Subsequently, we retain only the largest cluster $C_i^{max}$ as the refined candidate set and discard all others. As a result, semantic voting will only be applied to answers $a_i^j \in C_i^{max}$:

$$S_{SV}(a_i^j | C_i^{max}) = \frac{1}{|C_i^{max}|} \sum_{a_i^j, a_i^k \in C_i^{max}, k \neq j} f_{\text{sim}}(a_i^j, a_i^k) \tag{4}$$

---

**Algorithm 1:** Building DPO-style training pairs

**Input:** Base LLM $\mathcal{M}$; Unsupervised dataset $\mathcal{D}_u$; Sentence embedding model $\mathcal{M}_{emb}$; Sample size $N$; Clustering function $f_c$
**Output:** Training dataset for DPO $\mathcal{D}_{dpo}$

1 **for** $x_i$ *in* $\mathcal{D}_u$ **do**
2     Generate $N$ candidate answers; $\mathcal{A}_i = \{a_i^j | a_i^j \sim \mathcal{M}(\cdot | x_i)\}_{j=1}^N$;
3     Aggregate candidate answers into answer clusters; $\{C_i^k\}_k \leftarrow f_c(\{\mathcal{M}_{emb}(a_i^j) | a_i^j \in \mathcal{A}_i\}_j^N)$ ;
4     Get the maximum answer cluster; $C_i^{max} \leftarrow \arg\max_k |C_i^k|$;
5     Get the preferred answer; $a_i^w \leftarrow \arg\max_{a_i^j \in C_i^{max}} S_{SV}(a_i^j | C_i^{max})$ ;
6     Get the dispreferred answer; $a_i^l \leftarrow \arg\min_{a_i^j \in C_i^{max}} S_{SV}(a_i^j | C_i^{max})$ ;
7     Add $\{x_i, a_i^w, a_i^l\}$ to $\mathcal{D}_{dpo}$ ;
8 **end**

---

| | | wmt24pp_de | | wmt24pp_fr | | wmt24pp_ru | | wmt24pp_es | | cnn_dailymail | | pubmed | |
|---|---|---|---|---|---|---|---|---|---|---|---|---|---|
| | | BLEU | n-MQM | BLEU | n-MQM | BLEU | n-MQM | BLEU | n-MQM | RougeL | BLEURT | RougeL | BLEURT |
| Qwen-1.5B | base | 16.12 | 19.12 | 13.63 | 13.72 | 10.63 | 16.92 | 20.11 | 16.78 | 13.50 | 29.62 | 0 | 0 |
| | SJ | 3.45 | 9.16 | 6.73 | 9.95 | 4.06 | 9.35 | 1.94 | 7.47 | 13.54 | 27.46 | NA | NA |
| | EM | **18.21** | 20.20 | 19.77 | 19.49 | 11.67 | 18.45 | **24.32** | **20.16** | 1.63 | 4.41 | NA | NA |
| | SVSI | 18.04 | **20.53** | **20.34** | 19.96 | **11.93** | 18.48 | 23.46 | 19.88 | **13.72** | **30.63** | NA | NA |
| Llama-1B | base | 2.91 | 9.39 | 4.29 | 9.10 | 1.71 | 8.27 | 5.47 | 8.85 | 16.86 | 37.93 | 27.62 | 50.23 |
| | SJ | 2.06 | 9.09 | 2.81 | 8.33 | 1.98 | 8.64 | 2.57 | 7.78 | 16.53 | 38.02 | **28.27** | 50.54 |
| | EM | 1.51 | 8.68 | 4.35 | 8.80 | **4.86** | **11.11** | 5.21 | 8.65 | 14.45 | 25.66 | 28.14 | 45.69 |
| | SVSI | **5.28** | **10.60** | **5.59** | **9.74** | 2.75 | 9.10 | **6.89** | **9.35** | **17.83** | **40.28** | 27.70 | **50.92** |
| Qwen-3B | base | 17.64 | 20.12 | 19.69 | 18.83 | 11.90 | 17.06 | 23.71 | 19.12 | 0 | 0 | 0 | 0 |
| | SJ | 18.62 | 21.41 | 20.02 | 19.23 | 12.52 | 18.08 | 23.15 | 18.73 | NA | NA | NA | NA |
| | EM | 19.32 | 20.53 | 19.72 | 17.67 | **13.66** | 17.57 | 25.22 | 19.65 | NA | NA | NA | NA |
| | SVSI | **20.44** | **21.46** | **20.79** | **19.49** | 13.41 | **18.76** | **25.27** | **20.04** | NA | NA | NA | NA |
| Llama-3B | base | 6.35 | 10.85 | 8.20 | 10.46 | 3.63 | 9.67 | 10.03 | 10.17 | 7.23 | 16.25 | 27.77 | 49.91 |
| | SJ | 4.78 | 9.88 | 4.67 | 9.09 | 1.26 | 7.95 | 5.09 | 8.46 | 0.18 | 1.92 | 23.01 | 43.96 |
| | EM | **10.52** | **13.14** | **14.00** | **14.04** | **11.34** | **16.31** | **12.01** | **11.82** | **20.69** | **38.97** | **29.35** | 46.82 |
| | SVSI | 8.71 | 11.59 | 10.51 | 11.60 | 5.42 | 10.53 | 11.89 | 10.80 | 9.18 | 20.36 | 27.67 | **50.09** |
| Qwen-7B | base | 16.42 | 16.39 | 19.71 | 16.67 | 11.62 | 15.31 | 23.67 | 16.31 | 16.44 | 33.9 | 18.81 | 32.44 |
| | SJ | 18.34 | 18.45 | 21.20 | **18.52** | 12.08 | **16.34** | **25.58** | 17.70 | 7.49 | 16.33 | 23.91 | 38.95 |
| | EM | 7.32 | 10.05 | 17.66 | 14.18 | 11.63 | 14.95 | 17.01 | 11.48 | 17.65 | 33.75 | 29.63 | 46.1 |
| | SVSI | **20.19** | **19.12** | **21.76** | 17.76 | **12.35** | 16.29 | 25.53 | **17.76** | **20.69** | **42.61** | 29.63 | **47.76** |
| Llama-8B | base | 20.22 | 20.12 | 24.32 | 20.20 | 13.94 | 18.18 | 28.92 | 20.45 | 20.68 | 45.48 | 29.48 | 53.68 |
| | SJ | **20.83** | 20.62 | 11.46 | 12.29 | 7.19 | 12.02 | 26.08 | 18.12 | 0.08 | 1.73 | 10.34 | 21.08 |
| | EM | 0 | 0 | 22.35 | 19.96 | 8.98 | 12.85 | 12.4 | 9.01 | **23.23** | 44.78 | **33.86** | 51.62 |
| | SVSI | 20.76 | **20.92** | **24.74** | **20.66** | **14.54** | **19.12** | **29.33** | **20.45** | 20.72 | **45.53** | 29.69 | **54.03** |

Table 1: Evaluation of **SVSI** against Self-Judging (**SJ**) and Entropy Minimization (**EM**) on translation (BLEU, n-MQM) and summarization (Rouge-L, BLEURT) tasks; "NA" indicates cases where base models failed to generate valid outputs, rendering evaluation inapplicable.

## 3.3 Training

Equation 4 assigns a continuous score to each candidate answer, enabling a fine-grained ranking that reflects the relative response quality. In principle, these pseudo-signals can be leveraged by various training paradigms, for instance, as reward signals in online reinforcement learning or to build contrastive pairs in preference learning. However, such unsupervised signals inevitably contain noise, requiring a robust training framework to ensure consistent performance gains. To this end, we adopt the Direct Preference Optimization (DPO) (Rafailov et al., 2023) in SVSI, since recent studies (Chowdhury et al., 2024) have demonstrated that DPO exhibits provable robustness to mislabeled or imperfect data, rendering it especially well-suited to our setting. To maximize the preference margin, we construct training pairs by selecting the candidate with the highest voting score as the preferred response $a^w$ and the one with the lowest score as the dispreferred response $a^l$. The full procedure for generating DPO-style training pairs is formalized in Algorithm 1. Using the resulting dataset $\mathcal{D}_{dpo}$, we optimize the model with the standard DPO loss (Rafailov et al., 2023):

$$\mathcal{L}_{DPO} = -\mathbb{E}_{(x,a^w,a^l)\sim\mathcal{D}_{dpo}}[\log\sigma(\beta\log\frac{\pi_\theta(a^w|x)}{\pi_{ref}(a^w|x)} - \beta\log\frac{\pi_\theta(a^l|x)}{\pi_{ref}(a^l|x)})] \qquad (5)$$

where $\sigma$ denotes the sigmoid function, $\beta$ is the regularization coefficient, and $\pi_\theta$ and $\pi_{ref}$ represent the parameterized training policy and reference policy, respectively. We further discuss the applicability of semantic voting to alternative training paradigms in Section 4.5.

## 4 Experiments

### 4.1 Setup

#### 4.1.1 Datasets

We evaluate SVSI on two classical open-ended unverifiable NLP tasks: translation and summarization. For translation, we employed four subsets of WMT24++ (Deutsch et al., 2025), involving translating from German, French, Russian, and Spanish into English. These subsets are denoted as *wmt24pp_de*, *wmt24pp_fr*, *wmt24pp_ru*, and *wmt24pp_es* respectively. For summarization, we use the CNN/Dailymail dataset (*cnn_dailymail*) (Hermann et al., 2015), regarding news summarization, and the medical summarization dataset for PubMed abstracts (*pubmed*) (pieetie, 2025).

### 4.1.2 EVALUATION PROTOCOL

To enhance the comprehensiveness of the evaluation, we employ two kinds of metrics—lexical metric and semantic metric—measuring different aspects of the answers. For translation, we use BLEU (Papineni et al., 2002) as the lexical metric and MetricX-24 (Juraska et al., 2024) to assess semantic quality. MetricX-24 is a well-trained, reference-free evaluation model that provides automatic quality estimates in the form of raw MQM ratings, where lower scores indicate better translation quality. For consistency in interpretation, in the following sections, we report a normalized MQM score $n-MQM = \frac{1}{1+MQM}$, aligning the direction of improvement with that of other metrics.

For summarization, we adopt ROUGE-L (Lin, 2004) as the lexical metric and BLEURT (Sellam et al., 2020) as the semantic metric. We do not use reference-free evaluation methods for summarization, as recent studies (Zhang et al., 2024a) show that such methods often struggle to accurately assess outputs generated by large language models.

## 4.2 IMPLEMENTATION DETAILS

We use six base models of varying scales: three from the Llama series (Dubey et al., 2024) (1B, 3B, and 8B parameters) and three from the Qwen series (Yang et al., 2024) (1.5B, 3B, and 7B parameters). During self-generation, we sample 64 responses per input using a temperature of 0.7 and top_p of 0.9. For semantic voting, we use Sim-CSE (Gao et al., 2021) to obtain sentence embeddings and cluster generated responses via HDB-SCAN (Malzer & Baum, 2020). For evaluation, answers are generated via greedy decoding. Both self-generation and evaluation use identical zero-shot prompts, and all training hyperparameters are held constant across methods. Full implementation details are provided in Appendix B, and a sensitivity study about clustering algorithms can be found in Appendix C.

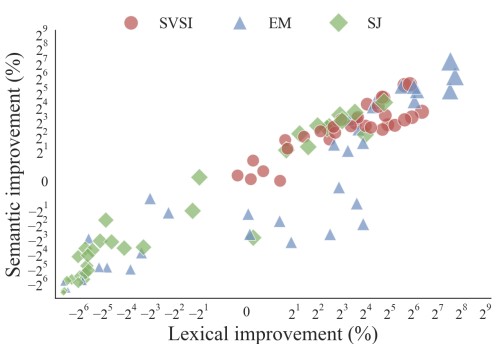

Figure 2: Overall lexical and semantic improvements of SVSI, EM, and SJ across all settings, relative to the base model performance.

## 4.3 MAIN RESULTS

In this section, we evaluate the effectiveness of **SVSI** through experiments conducted across all datasets and models. Three exceptions are excluded: Qwen-1.5B on *pubmed*, Qwen-3B on *cnn_dailymail*, and Qwen-3B on *pubmed*, where the base models fail to provide any valid answers. We compare against two self-evaluation baseline approaches: self-judging (**SJ**) and entropy minimization (**EM**). For SJ, we follow the setup from SRLM (Yuan et al., 2024), instructing the model to score its own generated responses on a 5-point scale criterion, and constructing preference pairs from the highest- and lowest-graded outputs. For EM, we adopt the trajectory-level entropy (Agarwal et al., 2025) as the measurement, selecting responses with the highest and lowest entropy for preference learning.

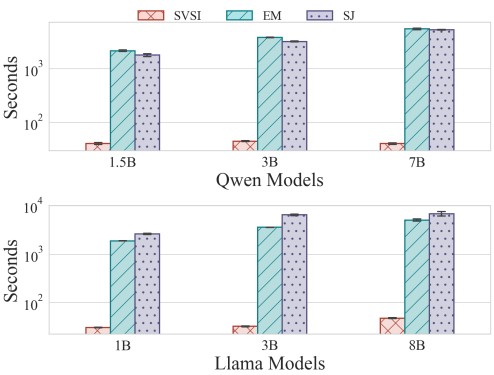

Figure 3: Computational overhead for building preference pairs via SVSI, EM, and SJ.

Table 1 presents the results. As shown, SVSI consistently outperforms the base model across all settings. Compared to the two self-evaluation baseline methods, SVSI performs on par with or better than them in most cases, though it is less effective in certain instances, particularly on the Llama-3B model. To provide an overall comparison, we compute the improvement percentage of each method relative to the base model across all configurations and visualize the results in Figure 2. The *x*-axis

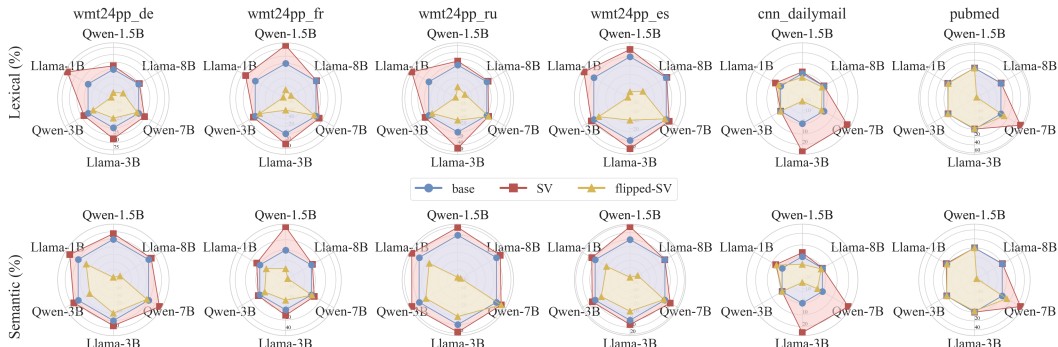

Figure 4: Improvements of training with Semantic Voting (**SV**) and **flipped-SV** (inverted SV preference pairs), relative to the base model on lexical (top row) and semantic (bottom row) metrics.

and *y*-axis represent improvements in lexical and semantic metrics, respectively, with logarithmic scaling applied to both axes. As the figure illustrates, while all three methods demonstrate significant gains in some cases, SVSI exhibits more consistent and stable performance, with no instances of severe degradation. This advantage likely stems from its independence from self-evaluation, thereby avoiding the pitfalls of model overconfidence.

Another notable advantage of SVSI is its significantly lower time overhead. We compute the average time required to construct preference pairs across all datasets for the three methods. The results, shown in Figure 3, reveal that SVSI requires orders of magnitude less time compared to the two baseline methods, and its computational cost does not scale with the size of the base model, making it particularly efficient for larger models.

## 4.4 VALIDATION FOR SEMANTIC VOTING

Recent studies (Shao et al., 2025) have shown that reinforcement learning with even a corrupted or spurious reward signal can significantly enhance reasoning capabilities in certain LLMs, perhaps because it triggers existing behaviors instilled during pretraining. This raises a concerning possibility that the surface effectiveness of newly designed reward signals may be just an illusion. Therefore, to rigorously assess the true contribution of semantic voting, we conduct experiments on a *flipped-SV* setting, where we invert the preference pairs constructed by semantic voting and proceed with DPO training using these reversed signals. If semantic voting really captures meaningful preferences, it can be anticipated that the *flipped-SV* would result in a performance degradation.

Results are presented in Figure 4, where the first and second row indicates improvements on lexical and semantic metrics, respectively, and each column corresponds to a different dataset. As shown, *flipped-SV* consistently underperforms standard semantic voting, and in nearly all cases also performs worse than the base model. In general, the extent of performance degradation in *flipped-SV* is proportional to the degree of improvement achieved by semantic voting, which aligns with our expectations. While there are a few isolated cases where *flipped-SV* still surpasses the base model, suggesting the presence of unreasonable improvement under spurious supervision as described in (Shao et al., 2025), this effect is too marginal to account for the primary gains observed; the main driver of improvements remains the valid signal captured by semantic voting.

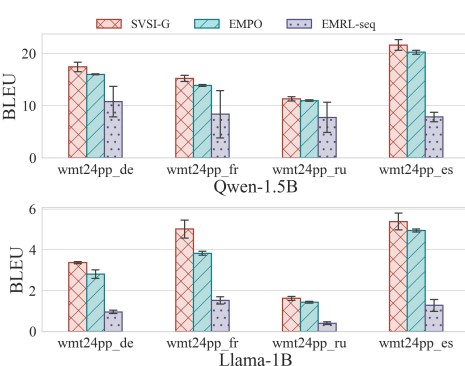

Figure 5: Evaluation of the GRPO-based variant of SVSI (SVSI-G) against EMPO and EMRL-seq.

## 4.5 EXTENSIBILITY ACROSS TRAINING METHODS

While we adopt DPO as our training framework in SVSI to improve robustness against noisy labels, semantic voting, in principle, assigns pseudo-labels to all self-generated responses, making it compatible with alternative training paradigms. In this section, we evaluate its extensibility to Group Relative Policy Optimization (GRPO) (Shao et al., 2024) by integrating it directly as the reward signal (denoted as **SVSI-G**). We compare against two GRPO-based baselines: **EMRL-seq**, which assigns rewards based on negative trajectory-level entropy (Agarwal et al., 2025), and **EMPO**, which first clusters generated samples and then rewards each response according to the size of the cluster it belongs to (Zhang et al., 2025b). Results are presented in Figure 5.

As shown, semantic voting with GRPO still demonstrates competitive average performance compared to the two baseline methods across four translation datasets. Nevertheless, both SVSI-G and EMRL-seq exhibit noticeable performance fluctuations across multiple runs, whereas EMPO shows markedly greater stability. We attribute this discrepancy to the form of reward signals. SVSI-G and EMRL-seq assign continuous scores to each sample, thereby constructing a preference ranking with high cardinality. However, due to the inherent noise in these pseudo-rewards, many preference pairs within the ranking are erroneous, which impedes stable convergence. In contrast, EMPO assigns discrete rewards and treats samples within the same cluster as equally preferred, trading off resolution for stability. This suggests a direction for future work: incorporating semantic voting into hybrid strategies that preserve the semantic richness while enhancing training stability.

## 4.6 ABLATION STUDY

In our proposed framework, we apply a clustering procedure to the self-generated responses before semantic voting. While this step is primarily intended to filter out interfering samples, thereby concentrating the candidate answers involved in voting, EMPO (Zhang et al., 2025b) has demonstrated that the clustering results also carry meaningful signals: samples in larger clusters tend to be of higher quality than those in smaller ones. To better understand the contribution of clustering in our approach, we design two ablation variants. The first, denoted as "*w/o. SV*", bypasses semantic voting and constructs preference pairs directly from the clustering results, selecting the preferred sample from the largest cluster and the dispreferred sample from the smallest. The second, "*w/o. clustering*", removes clustering and performs semantic voting directly over all self-generated samples. Results for both ablations are presented in Figure 6.

As shown, removing semantic voting consistently leads to a significant performance drop, indicating that clustering results alone can not provide sufficiently reliable signals for effective DPO training. In contrast, the impact of removing clustering varies across models. For Qwen-1.5B, performance remains stable, or even slightly improves, without clustering; while for Llama-1B, removing clustering consistently degrades performance. This discrepancy likely stems from the difference in base model capabilities: Llama-1B produces a higher proportion of noisy or off-topic responses, making clustering essential to ensure semantic voting operates on more coherent and meaningful samples. In contrast, Qwen-1.5B generates predominantly high-quality, low-noise responses during self-generation, rendering clustering redundant.

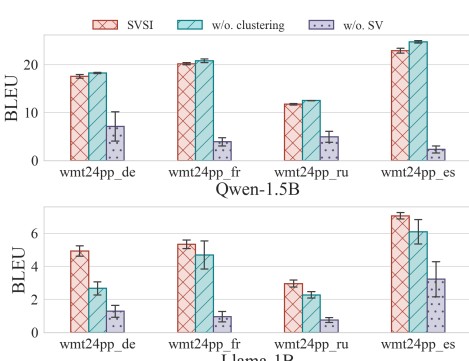

Figure 6: Ablation results for clustering and semantic voting (SV) in SVSI.

## 4.7 HYPERPARAMETER STUDY

In this section, we examine the influence of key hyperparameters on final performance across two critical stages of SVSI: the self-generation stage, where the *temperature* and *sampling size* jointly determine the candidate answer set (Figure 7); and the filtering stage, where parameters of the clustering algorithm govern the filtering results, thereby shaping the construction of preference pairs (Figure 8). All experiments are conducted on the *wmt24pp_de* dataset.

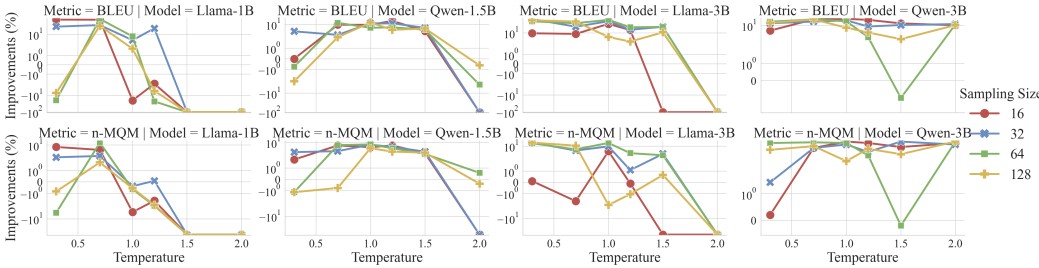

Figure 7: Relative improvements compared to base models across different generation parameters: *temperature* and the *sampling size*.

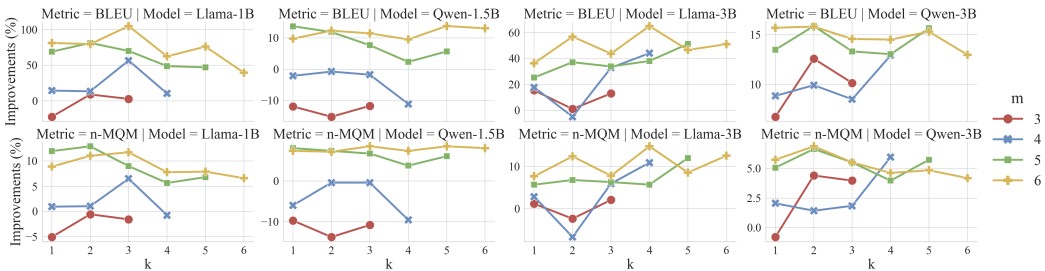

Figure 8: Relative improvements compared to base models across two different clustering parameters of HDBSCAN: minimum cluster size ($m$) and neighborhood-defining sample threshold ($k$).

Figure 7 presents the effects of *temperature* and *sampling size*. We evaluate four sampling sizes (ranging from 16 to 128) and six temperature values (from 0.3 to 2.0). As shown, larger sampling sizes generally yield more stable performance across varying temperatures. In contrast, smaller sampling sizes (*e.g.*, 16) introduce greater stochasticity into the candidate pool, resulting in increased performance fluctuation. As for temperature, values below 1.0 yield better results than higher ones. Excessively high temperatures tend to degrade response coherence, thereby disrupting cluster formation and potentially causing algorithmic failure. Model capacity also plays an important role: Qwen models demonstrate greater resilience to high temperatures than Llama models, and larger models overall exhibit improved stability compared to their smaller counterparts. Based on these observations, we recommend setting the temperature between 0.7 and 1.0, and using a sampling size of at least 32 to ensure reliable performance.

Figure 8 presents the impact of clustering hyperparameters on model performance. The clustering algorithm adopted in SVSI, HDBSCAN (Malzer & Baum, 2020), is governed by two key parameters: the minimum cluster size $m$ and the minimum number of samples required to define a neighborhood $k$. We evaluate four values of $m$ (ranging from 3 to 6), and for each, we test all valid $k$ values satisfying $k < m$. The results indicate that larger values of $m$ generally yield better overall performance. This can be attributed to that smaller $m$ values tend to fragment candidate responses into an excessive number of tiny clusters, thereby compromising clustering coherence and diminishing the number of semantically meaningful, well-structured groups. Moreover, when $m > 5$, performance becomes less sensitive to variations in $k$, suggesting enhanced robustness within this regime. In light of these findings, we recommend selecting larger $m$ values in practice to achieve more stable and effective clustering results.

## 4.8 SENSITIVITY STUDY ON EMBEDDING MODELS

In this section, we investigate the sensitivity of semantic voting to the choice of sentence embedding model. Specifically, we experiment with BGE-en (Xiao et al., 2023), DeBERTa-v3 (He et al., 2023), and the multilingual BGE-m3 (Chen et al., 2024), in addition to SimCSE, and summarize their downstream performance in Table 2.

| | | wmt24pp_de | | wmt24pp_fr | | wmt24pp_ru | | wmt24pp_es | | avg. | |
|---|---|---|---|---|---|---|---|---|---|---|---|
| | | BLEU | n-MQM | BLEU | n-MQM | BLEU | n-MQM | BLEU | n-MQM | BLEU | n-MQM |
| Qwen-1.5B | base | 16.12 | 19.12 | 13.63 | 13.72 | 10.63 | 16.92 | 20.11 | 16.78 | 15.12 | 16.63 |
| | DeBERTaV3 | 18.03 | 20.58 | 20.26 | 19.57 | 12.29 | 18.73 | 23.40 | 19.92 | 18.50 | 19.70 |
| | BGE-en | 17.98 | 20.70 | 19.97 | 19.65 | 12.13 | 18.21 | 22.36 | 19.19 | 18.11 | 19.44 |
| | BGE-m3 | 16.65 | 20.41 | 19.83 | 19.27 | 11.51 | 18.12 | 23.21 | 19.23 | 17.80 | 19.26 |
| | SimCSE | 18.04 | 20.53 | 20.34 | 19.96 | 11.93 | 18.48 | 23.46 | 19.88 | 18.44 | 19.71 |
| | cross-encoder | 18.35 | 20.62 | 19.82 | 19.53 | 11.92 | 18.80 | 23.14 | 19.69 | 18.31 | 19.66 |
| Llama-1B | base | 2.91 | 9.39 | 4.29 | 9.10 | 1.71 | 8.27 | 5.47 | 8.85 | 3.60 | 8.90 |
| | DeBERTaV3 | 3.11 | 9.26 | 4.48 | 9.26 | 1.54 | 7.96 | 5.01 | 8.55 | 3.54 | 8.76 |
| | BGE-en | 4.60 | 9.98 | 4.96 | 9.35 | 2.47 | 9.29 | 7.01 | 9.54 | 4.76 | 9.54 |
| | BGE-m3 | 5.15 | 10.37 | 4.19 | 8.96 | 1.47 | 8.24 | 7.66 | 9.92 | 4.62 | 9.37 |
| | SimCSE | 5.28 | 10.60 | 5.59 | 9.74 | 2.75 | 9.10 | 6.89 | 9.35 | 5.13 | 9.70 |
| | cross-encoder | 4.49 | 10.15 | 4.45 | 9.42 | 1.18 | 7.82 | 5.18 | 8.56 | 3.83 | 8.99 |

Table 2: Performance comparison among different semantic similarity computing approaches.

As the results indicate, all embedding models achieve nearly identical performance when applied to the Qwen-1.5B model. However, when applied to the Llama-1B model, DeBERTa-v3 exhibits a significant performance drop on certain tasks, leading to a notably lower overall score. We hypothesize that this discrepancy stems from the limited capacity of Llama-1B, which yields a less concentrated output distribution. This, in turn, amplifies variance at the embedding level across different models, rendering voting results more sensitive. Another noteworthy observation is that, despite being trained multilingually, BGE-m3 does not outperform monolingual embedding models, suggesting that multilingual capability is orthogonal to the effectiveness of semantic voting.

Beyond cosine similarity based on sentence embeddings, cross-encoders (Reimers & Gurevych, 2019) offer an alternative approach to measuring semantic similarity between text pairs. To assess their suitability within SVSI, we employ macro-TinyBERT [1]. The corresponding results are also reported in Table 2. Overall, cross-encoders perform comparably to embedding-based methods. Their main drawback lies in computational cost, necessitating $N^2$ forward passes since cross-encoders require pairwise inputs. Despite this limitation, we believe cross-encoders represent a promising avenue for future exploration, particularly when adapting SVSI to long-text scenarios.

## 5 CONCLUSION

In this paper, we propose semantic voting, a highly efficient, self-evaluation-free metric designed for self-improvement of large language models on unverifiable open-ended tasks. Instead of relying on computationally expensive and potentially biased self-evaluation methods, semantic voting leverages a lightweight sentence embedding model to encode self-generated responses and assigns pseudo-labels to each response based on its average semantic similarity with the others. Comprehensive experiments across various models and datasets demonstrate that semantic voting consistently matches or surpasses the performance of self-evaluation baselines, while reducing computational overhead by orders of magnitude (as low as merely 0.1%–5% of the original expense). Further analysis confirms that semantic voting offers a simple, effective, and scalable approach to facilitate large language models self-improvement.

## ACKNOWLEDGMENTS

This research was supported by Theme-based Research Scheme (T45-205/21-N) from Hong Kong RGC, and Generative AI Research and Development Centre from InnoHK.

## REPRODUCIBILITY STATEMENT

More implementation details are provided in Appendix B. All datasets used are publicly available, and the official code is released at https://github.com/rubickkcibur/Semantic-Voting.

---

[1] https://huggingface.co/cross-encoder/ms-marco-TinyBERT-L2-v2

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

APPENDIX

## A  THE USE OF LARGE LANGUAGE MODELS

In this work, we utilize large language models (LLMs) solely as writing assistants to identify grammatical errors and refine linguistic expressions for improved clarity and fluency.

## B  IMPLEMENTATION DETAILS

The concrete versions of Llama models we adopted in the experiments are: `Llama-3.2-1B-Instruct`, `Llama-3.2-3B-Instruct`, and `Llama-3-8B-Instruct`; and the versions of Qwen models are: `Qwen-2.5-1.5B-Instruct`, `Qwen-2.5-3B-Instruct`, and `Qwen-2.5-7B-Instruct`.

During the self-generation phase, for each input, we sample 64 responses using a temperature of 0.7 and top-p of 0.9. All generations are prompted in a zero-shot setting. To facilitate automated parsing, we instruct models to enclose their final answers within \boxed{}. Responses that fail to adhere to this formatting convention are excluded from preference pair construction. The prompt templates used for each task are illustrated in Figures 9, 10, and 11.

---

Prompt Template for *wmt24pp_de/fr/ru/es*

You are a translation assistant who carefully and thoughtfully translates German/French/Russian/Spanish sentences into English, ensuring that the translated sentences fluent and accurately convey the original meaning. Place your translation answer in \boxed{}. For axmaple, if the answer is "Hello World", you should output \boxed{Hello World}.

User:
The German/French/Russian/Spanish sentence is: '<source_sentence>'
Please think about how to translate step by step.

---

Figure 9: Prompt template for four *wmt24pp* translation tasks.

---

Prompt Template for *cnn_dailymail*

You are a skilled summarization assistant. Given a news report, provide a concise, informative summary in three sentences, totaling around 50 words. Please provide your summary in plain text and place it inside \boxed{}. For example, if your summary is "Hello World", you should output: \boxed{Hello World}

User:
Here is the news article:
<news_text>
Please summarize the article in three sentences.

---

Figure 10: Prompt template for the *cnn_dailymail* task.

For semantic voting, we employ SimCSE (Gao et al., 2021)[2] to obtain sentence embeddings and apply HDBSCAN (Malzer & Baum, 2020)[3] for clustering. Unless otherwise specified, we fix two key hyperparameters of HDBSCAN: the minimum cluster size ($m$) is set to 5, and the minimum number of samples required to define a neighborhood ($k$) is set to 2.

---

[2]https://huggingface.co/princeton-nlp/sup-simcse-bert-base-uncased
[3]https://pypi.org/project/hdbscan/

> **Prompt Template for _pubmed_**
>
> You are a skilled summarization assistant. Given a medical article, provide a concise, informative summary in one or two sentences, totaling around 50 words. Please provide your summary in plain text and place it inside \boxed{}. For example, if your summary is "Hello World", you should output: \boxed{Hello World}
>
> User:
> Here is the medical article:
> <medical_text>
> Please summarize the article in one or two sentences.

Figure 11: Prompt template for the _pubmed_ task.

| | | wmt24pp_de | | wmt24pp_fr | | wmt24pp_ru | | wmt24pp_es | | avg. | |
| | | BLEU | n-MQM | BLEU | n-MQM | BLEU | n-MQM | BLEU | n-MQM | BLEU | n-MQM |
|---|---|---|---|---|---|---|---|---|---|---|---|
| Qwen-1.5B | base | 16.12 | 19.12 | 13.63 | 13.72 | 10.63 | 16.92 | 20.11 | 16.78 | 15.12 | 16.63 |
| | MS | 18.54 | 20.28 | 20.59 | 19.76 | 12.23 | 18.59 | 24.79 | 20.04 | 19.04 | 19.67 |
| | OPTICS | 18.52 | 20.08 | 20.87 | 19.80 | 12.48 | 18.55 | 24.45 | 20.12 | 19.08 | 19.64 |
| | HDBSCAN | 18.04 | 20.53 | 20.34 | 19.96 | 11.93 | 18.48 | 23.46 | 19.88 | 18.44 | 19.71 |
| Llama-1B | base | 2.91 | 9.39 | 4.29 | 9.10 | 1.71 | 8.27 | 5.47 | 8.85 | 3.60 | 8.90 |
| | MS | 3.86 | 9.71 | 5.51 | 9.32 | 1.20 | 7.76 | 4.83 | 8.40 | 3.85 | 6.93 |
| | OPTICS | 4.34 | 9.75 | 6.20 | 9.83 | 2.06 | 8.22 | 7.32 | 9.13 | 4.98 | 9.23 |
| | HDBSCAN | 5.28 | 10.60 | 5.59 | 9.74 | 2.75 | 9.10 | 6.89 | 9.35 | 5.13 | 9.70 |

Table 3: Comparing performance among different clustering algorithms.

During DPO training, we use the AdamW optimizer (Loshchilov & Hutter, 2019) with a learning rate of $1 \times 10^{-6}$. The DPO regularization coefficient $\beta$ is set to 0.4. Training is conducted with a per-device batch size of 4 for a total of 4 epochs.

At evaluation, we generate responses using greedy decoding with a maximum output length of 512 tokens. An exception is made for `Qwen2.5-7B-Instruct`, whose reasoning traces tend to be substantially longer. To reduce parsing failures due to truncation, we extend the maximum generation length to 800 tokens for this model specifically.

All evaluation datasets are publicly available [4] [5] [6]. Due to computational constraints, we use a subset of $1,000$ samples for _cnn_dailymail_ and _pubmed_ datasets, rather than the full original sets. Codes for building these subsets are provided in the supplementary materials.

## C  Sensitivity study on Clustering algorithms

In this section, we evaluate the robustness of SVSI with respect to different clustering algorithms. In addition to HDBSCAN (Malzer & Baum, 2020), we employ two alternative density-based methods: OPTICS (Ankerst et al., 1999) and Mean Shift (MS) (Comaniciu & Meer, 2002). The corresponding results are shown in Table 3.

As shown in the table, OPTICS achieves performance nearly identical to that of HDBSCAN across both the Qwen-1.5B and Llama-1B models. In contrast, MS underperforms relative to these two methods on Llama-1B. This similarity between HDBSCAN and OPTICS is likely attributable to their shared foundation in the DBSCAN (Ester et al., 1996), which leads to analogous clustering behaviors. Nevertheless, despite the relatively weaker performance than the other two methods, MS still yields improvements over the base model. This underscores again the importance of applying a clustering-based filtering step to self-generated responses prior to semantic voting for less capable models, which is also demonstrated in Section 4.6.

---

[4]https://huggingface.co/datasets/google/wmt24pp

[5]https://huggingface.co/datasets/abisee/cnn_dailymail

[6]https://huggingface.co/datasets/pieetie/pubmed-abstract-summary

| | | wmt19_de | | wmt14_fr | | wmt19_ru | | wmt14_es | | avg. | |
|---|---|---|---|---|---|---|---|---|---|---|---|
| | | BLEU | n-MQM | BLEU | n-MQM | BLEU | n-MQM | BLEU | n-MQM | BLEU | n-MQM |
| Qwen-1.5B | base | 25.95 | 19.69 | 19.45 | 14.73 | 16.93 | 18.18 | 12.06 | 19.01 | 18.60 | 17.90 |
| | SVSI | 27.06 | 19.88 | 25.27 | 20.58 | 18.55 | 19.76 | 12.78 | 20.45 | 20.91 | 20.17 |
| Llama-1B | base | 9.02 | 9.29 | 11.03 | 9.99 | 4.36 | 7.97 | 4.83 | 9.15 | 7.31 | 9.10 |
| | SVSI | 11.96 | 10.70 | 9.64 | 9.74 | 6.06 | 9.10 | 5.10 | 9.43 | 8.19 | 9.74 |

Table 4: Out-of-distribution evaluation for SVSI on translation tasks.

## D  OUT-OF-DISTRIBUTION EVALUATION

In Section 4.3, we evaluate the effectiveness of semantic voting under a test-time setting, where the model demonstrates performance gains on the test set through unsupervised tuning. A natural follow-up is whether such improvements generalize to unseen, out-of-distribution data. To address this, we assess the improved models on unfamiliar datasets, without any adaptation, fine-tuning, or even unsupervised recalibration.

Specifically, we employ *wmt19_de*, *wmt19_ru*, *wmt14_fr*, and *wmt14_es* (Barrault et al., 2019; Bojar et al., 2014) to evaluate the models trained on *wmt24pp_de*, *wmt24pp_ru*, *wmt24pp_fr*, and *wmt24pp_es* correspondingly. Results are summarized in Table 4. As shown, although performance gains are not universal across all cases, semantic voting consistently improves results on the majority of unseen datasets. This demonstrates that the capability enhancement conferred by semantic voting is transferable beyond the original data.

## E  INSTRUCTION FOLLOWING EVALUATION

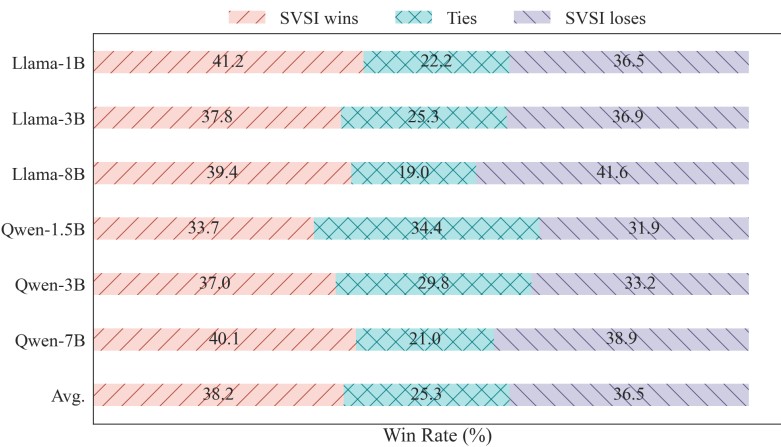

Figure 12: Instruction following ability improvement with SVSI.

In this section, we investigate the effectiveness of SVSI on a broader range of open-ended tasks. We evaluate its performance on AlpacaEval (Dubois et al., 2024), a benchmark for general instruction following, using DeepSeek-V3 (Liu et al., 2024) as the judge LLM. Specifically, we compare model performance before and after applying SVSI. The results, presented as win rates in Figure 12, demonstrate that SVSI can still yield general improvements across the instruction-following tasks.

## F  BACK-TRANSLATION EVALUATION

In Section 4, we evaluate translation tasks by translating multiple languages into English. Given the inconsistency of translation capabilities across different languages (Kocmi & Federmann, 2023; Jiang et al., 2025a), in this section, we investigate whether SVSI remains effective when applied to back-translation, *i.e.*, translating English into other languages. To support evaluation across a

| | | wmt24pp_de_inv | | wmt24pp_fr_inv | | wmt24pp_ru_inv | | wmt24pp_es_inv | |
|---|---|---|---|---|---|---|---|---|---|
| | | BLEU | n-MQM | BLEU | n-MQM | BLEU | n-MQM | BLEU | n-MQM |
| Qwen-7B | base | 12.18 | 16.53 | 17.44 | 14.45 | 6.51 | 13.07 | 20.39 | 14.75 |
| | SVSI | 15.22 | 20.58 | 19.19 | 15.72 | 8.75 | 15.11 | 22.98 | 16.67 |
| Llama-8B | base | 16.09 | 23.04 | 23.34 | 19.01 | 9.90 | 14.84 | 26.29 | 18.38 |
| | SVSI | 17.51 | 23.70 | 23.97 | 19.27 | 10.57 | 15.80 | 27.16 | 18.94 |

Table 5: Performance of translating English back to other languages.

| | | wmt24pp_de | wmt24pp_fr | wmt24pp_ru | wmt24pp_es | cnn_dailymail | pubmed |
|---|---|---|---|---|---|---|---|
| TTR | base | 86.36 | 86.17 | 86.01 | 86.04 | 79.61 | 86.86 |
| | SVSI | 86.08 | 86.09 | 85.85 | 85.78 | 78.16 | 85.56 |
| RSMD | base | 22.38 | 22.76 | 24.41 | 22.03 | 33.20 | 32.60 |
| | SVSI | 17.82 | 17.91 | 21.16 | 17.40 | 29.66 | 27.26 |

Table 6: Linguistic diversity changes after SVSI.

variety of languages, we replace SimCSE with BGE-M3 (Chen et al., 2024), a multilingually trained embedding model. As shown in Table 5, SVSI continues to deliver consistent improvements in this back-translation setting.

## G  SVSI-G DISCRETIZED VOTING SCORE

In Section 4.5, we explore the extensibility of our approach across training methods by integrating semantic voting into the GRPO algorithm, resulting in the variant SVSI-G. While SVSI-G achieves competitive performance than baseline methods, it exhibits fluctuations. We hypothesize that this instability stems from the nature of the reward signal. Specifically, the use of continuous scores leads to a high-cardinality ranking space. To test this hypothesis, we evaluate SVSI-G with coarse-grained rewards. Concretely, we discretize the original voting scores onto a 5-point scale by assigning the top 20% of scores to 5, the next 20% to 4, and so on. We refer to this discretized version as SVSI-G (discrete). As shown in Figure 13, SVSI-G (discrete) exhibits substantially reduced performance fluctuations compared to the original SVSI-G in Figure 5, which supports our hypothesis.

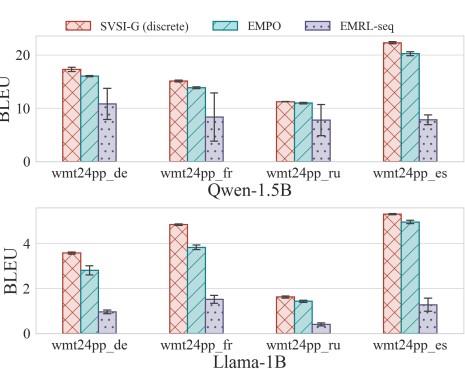

Figure 13: Evaluation of GRPO-based variant of SVSI with discretized voting score (SVSI-G-discrete).

## H  STUDY ON LINGUISTIC DIVERSITY

For open-ended tasks, beyond quality, creativity is also a crucial practical consideration. In this section, we examine how linguistic diversity is affected by SVSI. To quantify this, we employ the Type-Token Ratio (TTR) as a measure of lexical diversity and the root mean square deviation (RMSD) of semantic embeddings as a proxy for semantic diversity. Table 6 presents the results on Qwen-7B. As shown, SVSI leads to a reduction in linguistic diversity. In general, semantic diversity declines more substantially than lexical diversity, and the drop is more pronounced in translation tasks than in summarization tasks.

This observation aligns with an actively studied challenge in self-improvement frameworks, called model collapse (Guo et al., 2024; Shumailov et al., 2024; Alemohammad et al., 2023), where generative models, when trained iteratively on their self-generation, gradually narrow their output distri-

| | | wmt24pp_de | | wmt24pp_fr | | wmt24pp_ru | | wmt24pp_es | |
|---|---|---|---|---|---|---|---|---|---|
| | | BLEU | n-MQM | BLEU | n-MQM | BLEU | n-MQM | BLEU | n-MQM |
| Qwen-1.5B | SVSI | 18.04 | 20.53 | 20.34 | 19.96 | 11.93 | 18.48 | 23.46 | 19.88 |
| | SVSI-hn | 14.07 | 16.39 | 17.44 | 16.86 | 11.77 | 18.18 | 19.62 | 17.21 |
| Llama-1B | base | 5.28 | 10.60 | 5.59 | 9.74 | 2.75 | 9.10 | 6.89 | 9.35 |
| | SVSI-hn | 7.02 | 11.79 | 6.31 | 10.62 | 4.24 | 10.70 | 10.04 | 11.57 |

Table 7: Evaluation of using *hard negative* samples in SVSI.

| | | wmt24pp_de | wmt24pp_fr | wmt24pp_ru | wmt24pp_es |
|---|---|---|---|---|---|
| Qwen-7B | SV-ranking | 18.22 | 21.28 | 10.24 | 18.02 |
| | random-ranking | 1.55 | 1.25 | 0.46 | 0.56 |
| Llama-8B | SV-ranking | 23.42 | 23.88 | 17.69 | 24.02 |
| | random-ranking | 0.32 | 0.87 | 0.63 | 0.78 |

Table 8: Kendall's tau correlation of semantic voting-based ranking against reward-based ranking.

bution due to the absence of external information, eventually leading to incapability. However, we believe that this is unlikely to pose a serious practical concern, as multi-turn self-training of generative models is rarely employed in real-world settings. Moreover, if diversity in generated text is a primary objective, the simplest, most direct, and highly effective solution is to increase the sampling temperature.

## I STUDY ON UTILIZING HARD NEGATIVE SAMPLES

As discussed in Section 4.6, meaningful comparison signals across different clusters are difficult to obtain. However, certain *hard negative* samples, those responses with incorrect format, can be identified. While it is reasonable to assume that treating such samples as dispreferred examples and contrasting them with well-formatted responses could improve the model's adherence to formatting constraints, we deliberately excluded them in the standard SVSI formulation. By the way, we aim to eliminate confounding effects arising from formatting errors and to isolate the impact of semantic voting on genuine semantic quality.

In this section, we compare this design choice against an alternative that explicitly leverages *hard negative* samples. We denote this variant as SVSI-hn and evaluate it against the standard SVSI on Qwen-1.5B and Llama-1B. Results are presented in Table 7. As observed, incorporating *hard negative* yields better performance on Llama-1B but consistently underperforms on Qwen-1.5B. We hypothesize that this discrepancy arises because SVSI-hn concentrates the learning direction toward format compliance rather than semantic distinction. This bias can be beneficial for weaker models that frequently violate formatting instructions, but for more capable models like Qwen-1.5B, which already follow formatting reliably, such a focus may distract from deeper semantic refinement, ultimately degrading performance.

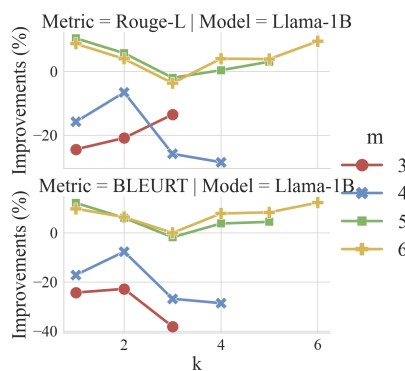

Figure 14: Hyperparameter study on cnn/dailymail.

## J ANALYSIS ON SEMANTIC VOTING-BASED RANKINGS

In this section, we assess the consistency between rankings derived from semantic voting scores and the true underlying quality. As a proxy for ground-truth quality, we employ the pre-trained translation reward model MetricX (Juraska et al., 2024), which has demonstrated strong alignment with human judgments. Specifically, we compute Kendall's tau coefficient between the ranking

induced by semantic voting scores and the ranking based on MetricX rewards. Table 8 presents the results. For reference, we also include the coefficient obtained from a random ranking. As shown, the semantic voting–based ranking exhibits a substantially higher correlation with MetricX than a random baseline.

# K HYPERPARAMETER STUDY ON CNN/DAILYMAIL

In Section 4.7, we examine how the hyperparameters of HDBSCAN affect final performance. As that section focuses only on translation tasks, we provide additional results on summarization. Specifically, Figure 14 shows the performance of Llama-1B on the CNN/DailyMail dataset. As can be seen, the conclusions drawn in Section 4.7 still hold: larger values of $m$ generally lead to better performance and greater robustness.

