# OpenReview forum: "Semantic Voting: A Self-Evaluation-Free Approach for Efficient LLM Self-Improvement on Unverifiable Open-ended Tasks"
_ICLR.cc/2026/Conference — ICLR 2026 Poster_

### Official Review · Reviewer_jSCx · 2025-10-21

**Soundness:** 2
**Presentation:** 3
**Contribution:** 2
**Rating:** 4
**Confidence:** 4

**Summary:**

This paper introduces Semantic Voting (SVSI), a novel self-improvement framework for LLMs on open-ended, unverifiable tasks. The core idea is to extend majority voting to the semantic space by generating multiple candidate responses, clustering them to find a semantic consensus, and then using the average semantic similarity as a score to construct preference pairs for DPO training. The method aims to be a computationally efficient and self-evaluation-free alternative to existing approaches.

**Strengths:**

1. SVSI offers a practical and computationally lightweight alternative to expensive self-evaluation methods (like self-judging) for unverifiable open-ended tasks.

2. The methodology is well-structured and clearly explained.

**Weaknesses:**

1. For unverifiable, open-ended tasks, diversity is a crucial metric. However, the proposed method may stifle diversity and creativity, as this unsupervised approach could lead to a severe Matthew effect. An answer that is creative, novel, or stylistically unique (yet still correct) is likely to be filtered out or receive a low voting score simply because its semantics differ from the 'mediocre' majority.

2. Its reliance on sentence embeddings makes its generalizability to long-text tasks questionable, where a single vector cannot capture full semantic complexity.

3. The experimental comparison is also limited, omitting more baselines and modern evaluation metrics like LLM-as-a-judge.

4. The cost analysis is potentially misleading; while cheaper than self-evaluation, the method's absolute cost is significantly higher than Supervised Fine-Tuning (SFT), yet no performance comparison is provided to justify this.

5. The pseudo-reward signal appears inherently unstable, as shown by its performance fluctuations with GRPO, suggesting its success may heavily depend on the noise robustness of the DPO framework.

**Questions:**

1. How does the method impact output diversity? Have you measured whether it causes a Matthew effect, leading to a less diverse distribution?

2. How can the method be adapted for long-text tasks where a single embedding is insufficient?

3. Were other unsupervised methods considered for comparison? For the tasks and models chosen in this paper, Supervised Fine-Tuning (SFT) would likely outperform SVSI with better cost-efficiency. Could you elaborate on the necessity of using SVSI in this context?

4. Can you explain the observed signal instability (e.g., with GRPO) and the method's sensitivity to its components, like the choice of embedding model?

---

> ### Author Response · Authors · 2025-11-20
> **Response to Reviewer jSCx (1)**
>
> Thanks for your valuable feedback. Here are some responses with clarification to misunderstandings and some new results.
>
> 1. About diversity. We do acknowledge that measuring diversity changes will help understand how output distribution shifts. And we also anticipate a diversity drop after training. For reference, we perform an additional investigation about linguistic diversity on Qwen-7B. We use TTR (Type-Token Ratio) to reflect lexical diversity and RSMD (root mean square deviation) among semantic embeddings to reflect semantic diversity. As shown, linguistic diversity decreases after SVSI. In general, semantic diversity decreases more than lexical diversity, and translation tasks drop more than summary.
>
>     | TTR   | wmt24pp_de | wmt24pp_fr | wmt24pp_ru | wmt24pp_es | cnn_dailymail | pubmed |
>     |-------|------------|------------|------------|------------|---------------|--------|
>     | base  | 86.36      | 86.17      | 86.01      | 86.04      | 79.61         | 86.86  |
>     | SVSI  | 86.08      | 86.09      | 85.85      | 85.78      | 78.16         | 85.56  |
>
>
>     | RSMD  | wmt24pp_de | wmt24pp_fr | wmt24pp_ru | wmt24pp_es | cnn_dailymail | pubmed |
>     |-------|------------|------------|------------|------------|---------------|--------|
>     | base  | 22.38      | 22.76      | 24.41      | 22.03      | 33.20         | 32.60  |
>     | SVSI  | 17.82      | 17.91      | 21.16      | 17.40      | 29.66         | 27.26  |
>
>     However, this may not be solely due to the voting machenism, but rather a common issue with all self-improvement methods. A recent study[1] comprehensively analyzed the impact of training LLMs on self-generated data on various linguistic diversity metrics and unfortunately found that the diversity decline is inevitable. This also meets findings in a broader study field called model collapse[2][3]. The core reason is also very intuitive: without external information, each self-improved output distribution can only be narrower and narrower with iteration. Moreover, we also believe that this trade-off is not a particularly serious issue, because in practice, if one truly seeks diversity in generated text, the simplest, most direct, and effective approach is to increase the sampling temperature, rather than modifying the training methodology.
>
> 2. About baselines and why not "LLM-as-a-judge". We have two unsupervised baselines in the experiments, one of which is just based on "LLM-as-a-judge". The EM represents entropy minimization, and the SJ represents self-judging. Both of these methods are unsupervised. Specifically, self-judging is based on "LLM-as-a-judge", where the model judges its own output. Section 4.3 shows more details.
>
> 3. About semantic voting on GRPO. As we discussed in section 4.5, the observed larger fluctuation than EMPO is largely attributed to the reward format. EMPO uses discrete rewards while semantic voting scores are continuous. Therefore, the solution is also intuitive, making the voting score coarse-grained. We try to discretize the voting score to 1-5 (mapping the top 20% to 5, the second top 20% to 4, and so on) instead of using the original numbers. The following are the results. As shown, the variance clearly decreases. It is worth noting that the absolute fluctuation of SVSI-G is not very significant. In most cases, even the bottom line of SVSI-G is beyond the topline of baseline methods.
> | methods | wmt24pp_de     | wmt24pp_fr     | wmt24pp_ru     | wmt24pp_es     |
> |-----------------|----------------|----------------|----------------|----------------|
> | SVSI-G-dense    | 17.43$\pm$0.74 | 15.23$\pm$0.48 | 11.33$\pm$0.33 | 21.63$\pm$0.85 |
> | SVSI-G-discrete | 17.63$\pm$0.17 | 15.11$\pm$0.13 | 11.24$\pm$0.03 | 22.30$\pm$0.17 |
>
> 4. For long-text tasks where a single embedding is insufficient. Another reviewer happened to mention replacing embedding models with cross-encoders, which is more adaptable to the text format. We perform an initial test of its effectiveness on current tasks. The following are the results. As shown, the cross-encoder performs comparably to SimCSE. We believe this may be a promising way to address long-term problems, even considering that the time efficiency of a cross-encoder is worse than that of sentence embeddings.
> | methods | wmt24pp_de | wmt24pp_fr | wmt24pp_ru | wmt24pp_es |
> |----------------------|------------|------------|------------|------------|
> | SVSI (SimCSE)        | 18.04      | 20.34      | 11.93      | 23.46      |
> | SVSI (cross-encoder) | 18.35      | 19.82      | 11.92      | 23.14      |

---

> > ### Author Response · Authors · 2025-11-20
> > **Response to Reviewer jSCx (2)**
> >
> > 5. About time-cost comparison to SFT. There may be some misunderstanding. Our context is LLM Self-improvement, where external supervision is not presumed. Therefore, SFT is not a valid baseline since it requires external annotations. Moreover, we also suspect that you might be asking why not replace the DPO part in SVSI with SFT. However, it is also kind of unreasonable, since regardless of the training method, the process of sampling self-generated responses and constructing pseudo-labels from them remains unchanged, and so does the time saved by semantic voting. We would appreciate it if you could elaborate more.
> >
> > **Reference**
> >
> > [1] Guo, Yanzhu, et al. "The curious decline of linguistic diversity: Training language models on synthetic text." Findings of NAACL. 2024.
> >
> > [2] Shumailov, Ilia, et al. "AI models collapse when trained on recursively generated data." Nature 631.8022 (2024): 755-759.
> >
> > [3]Alemohammad, Sina, et al. "Self-consuming generative models go mad." ICLR 2023.

---

> > ### Comment · Reviewer_jSCx · 2025-11-22
> > **Response to Authors**
> >
> > I think the author's response addresses most of my doubts and problems, especially on the impact of output diversity and the semantic voting of GRPO. For the problem of insufficient single embedding, the author also made a partial explanation and some results presentation, hoping to be included in the final version. So I rasied my score.

---

> > > ### Author Response · Authors · 2025-11-22
> > >
> > > Thank you very much for your recognition. We will incorporate the new results from our discussion into the final version.
> > >
> > > Once again, we sincerely appreciate your valuable feedback and the time you have dedicated.

---

### Official Review · Reviewer_F69H · 2025-10-21

**Soundness:** 4
**Presentation:** 3
**Contribution:** 3
**Rating:** 6
**Confidence:** 4

**Summary:**

This paper proposes Semantic Voting (SVSI), a lightweight and self-evaluation-free framework for improving large language models (LLMs) on unverifiable open-ended tasks such as translation and summarization. Unlike existing self-improvement methods (e.g., self-judging or entropy minimization), which rely on computationally costly and biased self-evaluation, SVSI replaces hard exact matching with soft semantic matching using sentence embeddings. The method clusters self-generated responses, applies semantic similarity-based voting, and uses the resulting pseudo-preference pairs for Direct Preference Optimization (DPO) training. Experiments across six models (Llama and Qwen series) and multiple datasets show that SVSI achieves comparable or superior performance to baselines while being 100× more computationally efficient.

**Strengths:**

- Introduces a new paradigm (semantic voting) that generalizes majority voting to open-ended tasks through soft semantic matching.

- Demonstrates a drastic reduction in computational cost (as low as 0.1–5% of self-evaluation methods) without sacrificing performance.

- Extensive evaluation across 6 models and multiple tasks with consistent gains in BLEU, ROUGE-L, and BLEURT.

- Includes flipped-pair tests, ablation studies, and hyperparameter analysis that strengthen methodological credibility.

- Algorithmic clarity and inclusion of implementation details in the appendix enhance replicability.

**Weaknesses:**

- The semantic similarity metric depends heavily on the choice of the sentence embedding model (SimCSE). Cross-model generalization is not deeply analyzed.

- Evaluations focus mainly on translation and summarization. It remains unclear how SVSI would perform on more subjective tasks (e.g., story writing or reasoning).

- Results rely solely on automatic metrics; human judgments would provide stronger validation of semantic quality improvements.

- As shown in Section 4.5, reward-based training variants exhibit performance fluctuation. The paper could provide further analysis on stabilization strategies.

**Questions:**

- How sensitive is SVSI to the choice of sentence embedding model? Would using multilingual or domain-specific encoders improve results?

- Could the method scale to even larger sampling sizes (e.g., 256 responses) without diminishing returns or excessive memory cost?

- Did you consider using model-based similarity scoring (e.g., cross-encoder) instead of embedding-based cosine similarity for higher fidelity?

- In the flipped-SV experiments, could the residual improvements over base models be due to pretraining biases or unintentional preference alignment?

- How would SVSI perform in reasoning or dialogue-generation tasks where “semantic similarity” may not correlate with quality?

**Details Of Ethics Concerns:**

The paper does not raise ethical or safety concerns. It uses publicly available datasets and standard open-source models. There is no use of human subjects or private data.

---

> ### Author Response · Authors · 2025-11-20
> **Response to Reviewer F69H (1)**
>
> Thanks for your constructive feedback. Here are some responses with clarification and some new results.
>
> 1. Embedding model generalization. We have an ablation study on embedding models in Appendix C, which shows that different embedding models yield similar performance on relatively capable models like Qwen, while varying on less capable models like Llama-1B, where SimCSE performs generally better. We put it in the appendix because of the page limitation. We will move it to the main content.
>
> 2. About multilingual embedding models. It's an interesting idea since we found that the three embedding models we used in Appendix C are all primarily trained on English corpus. Therefore, we try BGE-m3 (https://huggingface.co/BAAI/bge-m3) additionally. Here are the BLEU results on Qwen-1.5B. As it shows, although BGE-m3 can still achieve improvement, it is consistently less effective than SimCSE. This may suggest that when performing semantic similarity analysis on a specific language, multilingual embedding models are no (at least not significantly) better than models specifically trained for that language.
> | methods | wmt24pp_de | wmt24pp_fr | wmt24pp_ru | wmt24pp_es |
> |---------------|------------|------------|------------|------------|
> | SVSI (SimCSE) | 18.04      | 20.34      | 11.93      | 23.46      |
> | SVSI (BGE-m3) | 16.65      | 19.83      | 11.51      | 23.21      |
>
> 3. About using the cross-encoder as similarity measurement. It's also an interesting idea about semantic similarity computation. We try ms-marco-TinyBERT-L2-v2 (https://huggingface.co/cross-encoder/ms-marco-TinyBERT-L2-v2) to verify the effectiveness of cross-encoder models. Here are the results on Qwen-1.5B. As shown, using cross-encoder generally performs comparably to SimCSE, showing its promising effectiveness. The most concerning drawback of using cross-encoder is the computational cost. Obtaining embeddings requires only $N$ model forward passes, and the subsequent pairwise cosine similarity computations have negligible time cost. In contrast, a cross-encoder requires pairwise pairing at input time, necessitating $N^2$ forward passes. However, we do believe this is a promising direction to explore in the future, as model-based similarity computation is clearly more adaptable to the task.
> | methods | wmt24pp_de | wmt24pp_fr | wmt24pp_ru | wmt24pp_es |
> |----------------------|------------|------------|------------|------------|
> | SVSI (SimCSE)        | 18.04      | 20.34      | 11.93      | 23.46      |
> | SVSI (cross-encoder) | 18.35      | 19.82      | 11.92      | 23.14      |
>
> 4. About more tasks. We choose translation and summarization primarily for their well-established metrics (e.g., BLEU, Rouge-L), which provide a relatively objective and fair measurement. We acknowledge that experiments on more tasks would be beneficial. Therefore, we test SVSI on instructions from alpaca-eval, a dataset for general instruction following. We compare the responses before and after self-improvement, and use DeepSeek-V3 as the judge. The following are the results. As it shows, SVSI can still achieve general improvement over most of the models.
> | SVSI | Llama-1B | Llama-3B | Llama-8B | Qwen-1.5B | Qwen-3B | Qwen-7B | Avg.   |
> |------|----------|----------|----------|-----------|---------|---------|--------|
> | Win  | 41.25%   | 37.77%   | 39.38%   | 33.66%    | 37.02%  | 40.13%  | 38.20% |
> | Tie  | 22.23%   | 25.34%   | 19.01%   | 34.41%    | 29.81%  | 20.99%  | 25.30% |
> | Lose | 36.52%   | 36.89%   | 41.61%   | 31.93%    | 33.17%  | 38.88%  | 36.50% |
>
> 5. About human evaluation. We do not include human evaluation mainly because of its subjectivity and irreproducibility. While we do acknowledge that human evaluation provides valuable insights on unverifiable tasks, in practice, it is nearly impossible for us to organize professionals for a fair and comprehensive evaluation within such a short term. We'd like to provide an alternative assessment for your information. MetricX[1] is a well-established reward model evaluating multilingual translation quality, which embodies high consistency with human ratings as reported. We compute Kendall's tau coefficient between semantic-voting-based ranking with that ranked by MetricX, as an indirect reference to consistency with humans. The following are the results. As it shows, semantic-voting-based ranking has a much more significant correlation compared to a random ranking.
> | rankings | wmt24pp_de | wmt24pp_fr | wmt24pp_ru | wmt24pp_es |
> |---------------------------|------------|------------|------------|------------|
> | SV-ranking (Qwen-7B)      | 18.22      | 21.28      | 10.24      | 18.02      |
> | random-ranking (Qwen-7B)  | 1.55       | 1.25       | 0.46       | 0.56       |
> | SV-ranking (Llama-8B)     | 23.42      | 23.88      | 17.69      | 24.02      |
> | random-ranking (Llama-8B) | 0.32       | 0.87       | 0.63       | 0.78       |

---

> > ### Author Response · Authors · 2025-11-20
> > **Response to Reviewer F69H (2)**
> >
> > 6. About semantic voting with GRPO. As we discussed in section 4.5, although with fluctuation, SVSI-G consistently surpasses two baseline methods. We believe this fluctuation is not a very severe problem, since in most cases even the bottomline of SVSI-G is beyond the topline of baseline. For solutions, an intuitive way is to make the voting score coarse-grained. We discrete the voting score to 1-5 (mapping the top 20% to 5, the second top 20% to 4, and so on) instead of using the original numbers. The following are the results of Qwen-1.5B. As we can see, the variance clearly decreases with discrete rewards.
> > | methods | wmt24pp_de     | wmt24pp_fr     | wmt24pp_ru     | wmt24pp_es     |
> > |-----------------|----------------|----------------|----------------|----------------|
> > | SVSI-G-dense    | 17.43$\pm$0.74 | 15.23$\pm$0.48 | 11.33$\pm$0.33 | 21.63$\pm$0.85 |
> > | SVSI-G-discrete | 17.63$\pm$0.17 | 15.11$\pm$0.13 | 11.24$\pm$0.03 | 22.30$\pm$0.17 |
> >
> > 7. About larger sampling sizes. We discussed the influence of sampling size in Section 4.7. As Figure 7 shows, sampling size beyond 32 (32, 64, 128) already shows competitive performance and robustness to different temperatures. From 32 to 128, the improvements are quite marginal. So we believe scaling this number to 256 would also receive few benefits. As for the memory cost, the embedding models used are quite small, even compared to the smallest LLM (e.g., Llama-1B). Therefore, 256 would be no problem. Even when considering further increases in the sampling size or limitations in computational resources, batch computation of embeddings is still always feasible.
> >
> > 8. About residual improvements of the flipped-SV. It is quite complicated, especially considering this phenomenon does not happen universally. I recommend a recent study[1] on this problem, where the authors found in some cases, even a random reward can improve pretrained LLMs. In our opinion, this may be related to data leakage during pretraining. The model may just remember some knowledge obtained from a pretrained corpus during adaptation. Another possible explanation is that this residual improvement stems from a reduction in the model's outputs that violate the required format, since we only use format-compliant responses to construct training pairs (both preferred and dispreferred). Our flipped study was just designed for this worrying possibility. And it does prove that the improvement (at least quite a part of the improvement) of SVSI comes from the signal constructed by semantic voting. While the investigation into this phenomenon is really interesting, it seems orthogonal to the main thread of semantic voting. Therefore, we will not delve too much into it in this paper.
> >
> > **Reference**
> >
> > [1] Shao, Rulin, et al. "Spurious rewards: Rethinking training signals in rlvr." arXiv preprint arXiv:2506.10947 (2025).

---

### Official Review · Reviewer_ikCz · 2025-10-31

**Soundness:** 1
**Presentation:** 2
**Contribution:** 1
**Rating:** 2
**Confidence:** 4

**Summary:**

This paper focuses on addressing the limitations of existing LLM self-improvement methods for unverifiable open-ended tasks (e.g., machine translation, text summarization), such as high computational overhead and overconfidence bias in self-evaluation approaches (self-judging, entropy minimization) and the inapplicability of majority voting (relying on exact matching) to such tasks. It proposes the
Semantic Voting-based Self-Improvement (SVSI)
framework, a self-evaluation-free solution.

SVSI draws inspiration from majority voting but relaxes hard exact matching to soft semantic similarity matching. The framework first enables an LLM to generate multiple candidate responses for a given input,then uses a lightweight sentence embedding model (e.g., SimCSE) to encode these responses into semantic vectors. After filtering outliers via density-based clustering (HDBSCAN) and retaining only the largest cluster to ensure meaningful semantic consensus, it calculates each response’s "voting score" as the average cosine similarity with other responses in the cluster. The highest and lowest scoring responses form preference pairs, which are used to fine-tune the LLM via Direct Preference Optimization (DPO).

Empirical evaluations across 6 models (Qwen series: 1.5B, 3B, 7B; Llama series: 1B, 3B, 8B) and 6 datasets (4 WMT24++ translation subsets, CNN/DailyMail and PubMed summarization datasets) show that SVSI consistently outperforms self-judging (SJ) and entropy minimization (EM) baselines in most cases, with computational overhead only 0.1%–5% of self-evaluation methods. Additional validations (e.g., flipped-SV experiments to confirm valid semantic signals, out-of-distribution tests to verify generalization) and ablation studies (on clustering and embedding models) further confirm SVSI’s effectiveness, stability, and adaptability.

**Strengths:**

Targeted Problem-Solving for Unverifiable Tasks:
The paper addresses a critical gap in LLM self-improvement—existing methods either rely on computationally expensive, bias-prone self-evaluation (e.g., self-judging, entropy minimization) or hard-matching majority voting (inapplicable to open-ended tasks like translation and summarization). Its proposed Semantic Voting-based Self-Improvement (SVSI) framework avoids self-evaluation entirely, enabling lightweight and effective optimization for unverifiable open-ended tasks .

Innovative and Efficient Core Mechanism
:
Inspired by majority voting but extended to "soft semantic matching," SVSI uses lightweight sentence embedding models (e.g., SimCSE) to quantify semantic similarity, replacing rigid exact matching. This design drastically reduces computational overhead—its cost is only 0.1%–5% of self-evaluation baselines—while maintaining effectiveness, making it scalable even for larger models .

Practical Engineering Value
:
The framework is operationally transparent (detailed implementation in Appendix B) and requires no complex dependencies—self-generation uses standard sampling, embedding/clustering rely on lightweight tools, and DPO training uses standard hyperparameters. This makes SVSI easy to reproduce and deploy in real-world LLM optimization pipelines .

**Weaknesses:**

The proposed semantic voting approach relies on a straightforward pipeline: clustering self-generated responses using semantic similarity and calculating voting scores via average cosine similarity between sentence embeddings. This core logic lacks impactful technical novelty, as both semantic clustering and similarity-based ranking are well-established techniques in NLP. Notably, the ablation study confirms that clustering is not universally beneficial—for high-capability models like Qwen-1.5B, removing clustering either preserves performance or slightly improves it, which further underscores the framework’s simplicity and its limited innovative contribution to solving challenges in unverifiable open-ended tasks .

Weak Connection in Related Works
:
The related works section only critiques limitations of existing methods (e.g., majority voting’s reliance on exact matching, self-evaluation’s high computational overhead) but fails to link these flaws to specific components of the proposed semantic voting. It does not clarify how modules like lightweight sentence embedding or soft semantic matching directly address the drawbacks of prior approaches, weakening the rationale for the new method.

Disordered Section Reference in Figures
:
Figure 1, which outlines the SVSI framework, references Section 3.2 (self-generation filtering via clustering) before Section 3.1 (semantic voting)—a misalignment with the logical order of the Methodology chapter, causing confusion about the workflow sequence .

Unjustified Inclusion of Non-Original equation
:
Eq. 5 (pertaining to DPO loss) is directly adopted from prior work (Rafailov et al., 2023) without modifications. The paper provides no reason for including this non-innovative equation in the Methodology section, as it is irrelevant to the core semantic voting mechanism.

Ambiguous Figure 4
:
Figure 4 (comparing semantic voting and flipped-SV) lacks a clear legend. Key elements—such as labels for "SV" and "flipped-SV" or detailed axis annotations for lexical/semantic metrics—are missing, making it impossible to accurately interpret performance differences between the two settings .

Unjustified Clustering Algorithm Selection
:
The paper uses HDBSCAN for clustering self-generated responses but offers no experimental, theoretical, or literature-based justification for this choice. It does not compare HDBSCAN with other common clustering algorithms (e.g., DBSCAN, OPTICS) to demonstrate its suitability for semantic voting .

Sensitive and Opaque HDBSCAN Parameters
:
HDBSCAN’s performance depends on two key parameters (minimum cluster size m and neighborhood sample threshold k), which are shown to be sensitive to task and model variations (e.g., larger m improves performance for most models). However, the paper does not discuss whether these parameters require task-specific tuning or provide guidelines for adapting them to different unverifiable tasks, increasing practical application complexity .

Unjustified Metric Choices
:
For evaluation, the paper uses different metric-model pairs across tasks (e.g., BLEU/n-MQM for translation, ROUGE-L/BLEURT for summarization) but provides no explanation for these selections. It fails to justify why these metrics are optimally suited for measuring output quality in translation vs. summarization, weakening result validity .

Limited Task and Domain Coverage
:
The method is designed for unverifiable open-ended tasks but only tested on translation and summarization. No experiments check for potential negative effects in closed-ended tasks (e.g., arithmetic, multiple-choice), which is critical for assessing generalizability. The narrow task scope also makes it hard to confirm effectiveness across diverse unverifiable scenarios .

No Significant Advantage Over Baselines
:
In Table 1 and Figure 2, SVSI performs on par with or slightly better than baselines (SJ, EM) but shows no substantial, consistent advantage. For example, on Llama-3B, EM outperforms SVSI on multiple translation datasets (e.g., wmt24pp de BLEU: 10.52 vs. 8.71), failing to demonstrate value beyond existing methods .

Stability Issues in GRPO Adaptation
:
As noted in Section 4.5, when adapted to GRPO (SVSI-G), the method exhibits severe performance fluctuations across runs—far less stable than EMPO. This undermines confidence in its applicability beyond DPO training .

Unproven Clustering Necessity
:
The ablation study (Section 4.6) shows clustering is unnecessary for high-capability models like Qwen-1.5B (removing it slightly improves performance) and only beneficial for low-capability models like Llama-1B. This indicates the clustering component— a core part of SVSI—lacks universal necessity .

**Questions:**

1.Given that the core logic of semantic voting (semantic clustering + similarity-based scoring) is relatively simple and lacks significant technical innovation, and the ablation study shows clustering is not universally useful (e.g., it does not harm Qwen-1.5B’s performance when removed), how does the paper justify the novelty of the proposed framework compared to existing semantic similarity-based or clustering-based methods in LLM self-improvement?

2.In the related works section, the paper only critiques the limitations of existing methods (e.g., majority voting’s reliance on exact matching, self-evaluation’s high overhead) but fails to link these flaws to specific modules of semantic voting (e.g., lightweight sentence embedding, soft matching). Why does the paper not explicitly explain which module addresses which drawback of prior work, and how to verify the causal relationship between these modules and the solved problems?

3.Figure 1 (the SVSI framework overview) references Section 3.2 (clustering-based filtering) before Section 3.1 (semantic voting), which contradicts the logical order of the Methodology chapter. What is the reason for this disordered section reference, and does it reflect a misunderstanding of the framework’s own workflow?

4.Eq. 5 (the DPO loss formula) is directly adopted from Rafailov et al. (2023) without any modifications,
yet it is included in the Methodology section. Since this equation is not an original contribution and is
irrelevant to the core semantic voting mechanism, why does the paper choose to present it, and what
value does it add to explaining the proposed method?

5.Figure 4 (comparing semantic voting and flipped-SV) lacks a clear legend, making it impossible to
accurately identify which curve represents "SV" and which represents "flipped-SV," and unclear how
the axes correspond to lexical/semantic metrics. Why is such a critical result figure missing key
annotation information, and how does this affect the credibility of the conclusion that "flipped-SV
underperforms"?

6.The paper selects HDBSCAN as the clustering algorithm for filtering self-generated responses but
provides no experimental comparison (e.g., with k-means, DBSCAN) or theoretical justification (e.g.,
alignment with semantic voting’s needs) for this choice. What evidence supports that HDBSCAN is
more suitable for semantic voting than other common clustering algorithms?

7.HDBSCAN’s performance depends on sensitive parameters (minimum cluster size m and
neighborhood threshold k), and the paper shows larger m generally improves performance. However,
it does not explain whether these parameters need to be tuned for different tasks (e.g., translation vs.
summarization) or models. How should users determine the optimal m and k values when applying
SVSI to new unverifiable tasks?

8.For evaluation, the paper uses different metric pairs (BLEU/n-MQM for translation, ROUGE-L/BLEURT
for summarization) but provides no explanation for these choices. Why are these specific metrics
selected instead of others (e.g., using BLEURT for translation or n-MQM for summarization), and how
to ensure the selected metrics can fairly reflect the method’s performance across tasks?

9.The method is designed for unverifiable open-ended tasks but only tested on translation and
summarization. No experiments are conducted to check if SVSI causes negative effects (e.g., reduced
accuracy) in closed-ended tasks (e.g., arithmetic, multiple-choice). Why is the scope of experimental
verification so narrow, and how to confirm the method’s safety when applied to mixed task scenarios?

10.In Table 1, SVSI shows no significant advantage over baselines (e.g., on Llama-3B’s wmt24pp de task,
EM’s BLEU score is 10.52 while SVSI’s is 8.71). The paper claims SVSI is "effective," but why does it
not address the cases where it underperforms baselines, and how to prove its superiority in practical
application scenarios?

11.Section 4.5 notes that when SVSI is adapted to the GRPO framework (SVSI-G), it exhibits obvious
performance fluctuations, which are more severe than EMPO’s stability. Since the paper emphasizes
SVSI’s "efficiency and robustness," why does it not solve the stability issue in alternative training
paradigms, and what limitations does this reveal about the method’s adaptability?

12.The ablation study (Section 4.6) shows clustering is unnecessary for high-capability models like
Qwen-1.5B (removing it slightly improves performance) and only beneficial for low-capability models
like Llama-1B. As a core component of SVSI, why does clustering lack universal necessity, and how to
adjust the framework to avoid redundant steps for high-capability models?

13.SVSI relies on sentence embedding models (e.g., SimCSE) to calculate semantic similarity. If the
embedding model fails to capture task-specific semantics (e.g., medical terminology in PubMed
summarization), the pseudo-label quality declines. How does the paper plan to address the
dependence on embedding model performance, and is there a way to reduce the method’s sensitivity
to embedding model selection?

14.Semantic similarity metrics cannot distinguish between "semantically consistent but low-quality" and
"semantically distinct but high-quality" responses, leading to "conformity bias" (prioritizing majority
consensus over individual accuracy). Does the paper acknowledge this limitation, and what
improvements can be made to avoid discarding accurate but niche responses?

---

> ### Author Response · Authors · 2025-11-20
> **Response to Reviewer ikCz (1)**
>
> Thanks for your valuable time for providing such detailed feedback. We find many of the concerns may stem from misunderstandings. We hope the following responses will clarify them.
>
> **Concerns on paper writing**
> 1. About the section reference in Figure 1. Actually, we are afraid that we can not agree that this is a problem. The aim of Figure 1 is to show the overview of the framework; thus, it is very appropriate to describe in the order of data flow. In contrast, the Methodology section generally discusses design philosophy (i.e., not only what we did but also why we did). Therefore, it is necessary to first present semantic voting and discuss clustering later because the reason why we need clustering (as we discussed in Section 3.2) is to make semantic voting reliable for less capable models. Although you believe this "reflects a misunderstanding of the framework’s own workflow", on the contrary, we did this precisely to align with the design logic and enhance readability. After all, the figure caption is not the catalog.
>
> 2. About the related work. Just as you mentioned, the concept pairs, exact matching *v.s.* soft matching and high-overhead *v.s.* Lightweight, already have quite intuitive links. In paragraph 4 and 5 of Section 1, we clearly describe the sufferings of self-evaluation methods and how semantic voting avoids these problems by bypassing self-evaluation. For verification, this is exactly why there are Section 4.3 and Section 4.4. If your suggestion is to repeat this information in Section 2, we can do that since it is effortless. But we think it is less necessary considering the page limitation.
>
> 3. About the DPO formula. We present the DPO formula for enhancing readability. We do not presume all readers know the DPO formula well. For them, it is always more convenient to see the final loss directly than to refer to the original paper. Of course, we clearly claimed "we optimize the model with the standard DPO loss" with proper citation. Actually, this is not a rare practice[1][2][3]. Nevertheless, removing this formula is always with ease.
>
> 4. About Figure 4. Actually, just located in the tight center of the figure is the legend. And as the figure caption presents, the top row reflects lexical metrics and the bottom row reflects semantic metrics.
>
> **Concerns on experiments**
>
> 5. About clustering algorithm ablation. We have an ablation study about the clustering algorithm in Appendix D, where we compare HDBSCAN with OPTICS and Mean Shift. We put it in the appendix because of the page limitation. We will move it to the main content.
>
> 6. About HDBSCAN's hyperparameters. We acknowledge that performing hyperparameter studies on more tasks would be beneficial. We conduct additional experiments of Llama-1B on CNN/dailymail dataset. Settings are the same as in Figure 8. Here present the results (each column corresponds to the same k and each row corresponds to the same m). For simplicity, only Rouge-L is reported since the trend of BLEURT is identical. As shown, the conclusion, larger m generally improves performance, remains unchanged.
> | m\k | 1     | 2     | 3     | 4     | 5     | 6     |
> |-----|-------|-------|-------|-------|-------|-------|
> | 3   | 12.75 | 13.35 | 14.59 |       |       |       |
> | 4   | 14.20 | 15.76 | 12.51 | 12.07 |       |       |
> | 5   | 18.62 | 17.83 | 16.53 | 16.93 | 17.38 |       |
> | 6   | 18.33 | 17.53 | 16.25 | 17.54 | 17.52 | 18.46 |
>
> 7. About metric selection. The metrics are selected mainly based on the tasks they were originally designed for. BLEU[4] and MQM[5] are designed for machine translation. Rouge-L[6] is designed for summarization. BLEURT is applied to both tasks. And we use it for summarization because studies show BLEURT yields generally the best correlation with human scores among various automatic metrics[7].
>
> 8. Why not close-ended tasks. Just as you said, "The method is designed for unverifiable open-ended tasks". For closed-ended tasks, there is no need to do "soft matching". We recommend hard matching majority voting on these tasks, which has been proven to be simple yet effective[8][9]. However, we acknowledge that experiments on more open-ended tasks would be beneficial. Therefore, we test SVSI on instructions from alpaca-eval, a dataset for general instruction following. We compare the responses before and after self-improvement, and use DeepSeek-V3 as the judge. The following are the results. As it shows, SVSI can still achieve general improvement over most of the models.
> | SVSI | Llama-1B | Llama-3B | Llama-8B | Qwen-1.5B | Qwen-3B | Qwen-7B | Avg.   |
> |------|----------|----------|----------|-----------|---------|---------|--------|
> | Win  | 41.25%   | 37.77%   | 39.38%   | 33.66%    | 37.02%  | 40.13%  | 38.20% |
> | Tie  | 22.23%   | 25.34%   | 19.01%   | 34.41%    | 29.81%  | 20.99%  | 25.30% |
> | Lose | 36.52%   | 36.89%   | 41.61%   | 31.93%    | 33.17%  | 38.88%  | 36.50% |

---

> > ### Author Response · Authors · 2025-11-20
> > **Response to Reviewer ikCz (2)**
> >
> > 9. About underperforming cases. Although SVSI does not achieve the best performance in all cases, it still has a generally better (at least comparable) effectiveness, as shown in Figure 2. And underperforming than EM in Llama-3B does not mean ineffectiveness, as SVSI still brings consistent improvement compared to the base model, which means "effective" self-improvement.
> >
> > 10. About semantic voting on GRPO. As we discussed in section 4.5, although with fluctuation, SVSI-G consistently surpasses two baseline methods. We believe this fluctuation is not a very severe problem, since in most cases even the bottomline of SVSI-G is beyond the topline of baseline. For solutions, an intuitive way is to make the voting score coarse-grained. We discrete the voting score to 1-5 (mapping the top 20% to 5, the second top 20% to 4, and so on) instead of using the original numbers. The following are the results of Qwen-1.5B. As we can see, the variance clearly decreases with discrete rewards.
> > | methods | wmt24pp_de     | wmt24pp_fr     | wmt24pp_ru     | wmt24pp_es     |
> > |-----------------|----------------|----------------|----------------|----------------|
> > | SVSI-G-dense    | 17.43$\pm$0.74 | 15.23$\pm$0.48 | 11.33$\pm$0.33 | 21.63$\pm$0.85 |
> > | SVSI-G-discrete | 17.63$\pm$0.17 | 15.11$\pm$0.13 | 11.24$\pm$0.03 | 22.30$\pm$0.17 |
> >
> > 11. About the need for clustering. As presented in Section 3.2, clustering is to facilitate the stability of self-improvement by preventing significantly diverging candidates from undermining the effectiveness of semantic voting, particularly for less capable models. That is to say, the clustering is just designed to enhance the adaptability on less capable models. And the ablation study in Section 4.6 confirms, rather than diminishes, its necessity. In the less capable model, Llama-1B, a significant performance drop is observed when clustering is removed. Adjustment for high-capability models is also quite intuitive. Remove the clustering step.
> >
> > 12. About dependence on embedding models. In the current framework, embedding models are necessary to compute semantic similarities. For more professional tasks, we anticipate that a more specific model, such as a model trained on the professional corpus, can be used to represent semantic meanings. Since this exploration direction is orthogonal to this work, we would leave it for future work.
> >
> > 13. About conformity bias. It is indeed a limitation of voting-based signals, not only our semantic voting but also those based on exact matching[8][9]. And just as we discussed in Section 2.2 and Section 2.3, unsupervised or self-supervised signals inevitably inherent different types of bias. Therefore, it is really non-trivial to overcome this under unsupervised constraints. In practice, a more feasible way may be combining self-improvement with minimal human annotation guidance, leveraging techniques like importance weighting[10]. In this way, the bias in self-improvement can be shifted without involving too much human effort, which is more suitable for practical scenarios.
> >
> > **Concerns on novelty**
> >
> > 14. While we respect the reviewer's right to have subjective opinions on "what is novelty", we are afraid we can not approve judging a work to be lacking novelty because "the core logic is relatively simple" and "the techniques are well-established". On the contrary, we would recognize these features as a compliment, because they indicate that the method is easy to understand and reproduce. Meanwhile, we'd like to share two impactful papers with you. One is classical[11] and the other[12] is trendy. Both of their core mechanisms are very simple and are not firstly proposed.
> >
> > We hope our responses help clarify some misunderstandings.

---

> > > ### Author Response · Authors · 2025-11-20
> > > **Response to Reviewer ikCz (3)**
> > >
> > > **Reference**
> > >
> > > [1]SimPO: Simple Preference Optimization with a Reference-Free Reward. Yu Meng, Mengzhou Xia, Danqi Chen. NIPS 2024
> > >
> > > [2]Yu, Tianyu, et al. "Rlhf-v: Towards trustworthy mllms via behavior alignment from fine-grained correctional human feedback." CVPR 2024.
> > >
> > > [3]Wang, Junxiong, et al. "The mamba in the llama: Distilling and accelerating hybrid models." NIPS 2024
> > >
> > > [4]Papineni, Kishore, et al. "Bleu: a method for automatic evaluation of machine translation." ACL 2002.
> > >
> > > [5]Juraska, Juraj, et al. "MetricX-23: The Google submission to the WMT 2023 metrics shared task." Proceedings of the Eighth Conference on Machine Translation. 2023
> > >
> > > [6]Lin, Chin-Yew. "Rouge: A package for automatic evaluation of summaries." Text summarization branches out. 2004.
> > >
> > > [7]Zhang, Tianyi, et al. "Benchmarking large language models for news summarization." TACL 2024.
> > >
> > > [8] Huang, Jiaxin, et al. "Large language models can self-improve." EMNLP. 2023.
> > >
> > > [9] Zuo, Yuxin, et al. "Ttrl: Test-time reinforcement learning." arXiv preprint arXiv:2504.16084 (2025).
> > >
> > > [10]Jiang, Chunyang, et al. "Importance weighting can help large language models self-improve." AAAI 2025.
> > >
> > > [11]He, Kaiming, et al. "Deep residual learning for image recognition." CVPR 2016.
> > >
> > > [12]Wang, Xuezhi, et al. "Self-Consistency Improves Chain of Thought Reasoning in Language Models." ICLR 2023.

---

> > > ### Comment · Reviewer_ikCz · 2025-11-28
> > >
> > > The authors have addressed my concern. I will raise my score to 6.

---

### Official Review · Reviewer_nFpD · 2025-11-01

**Soundness:** 3
**Presentation:** 3
**Contribution:** 3
**Rating:** 6
**Confidence:** 4

**Summary:**

The paper introduces Semantic Voting–based Self‑Improvement (SVSI) for tasks without verifiable answers. The method samples multiple answers, embeds them, and filters them down to the largest semantic cluster. Within this cluster, it ranks answers by their average cosine similarity to all other answers. The top-ranked ("winner") and bottom-ranked ("loser") answers are then used as a preference pair for DPO training. This approach uses a simple sentence encoder and clustering, avoiding expensive LLM-as-a-judge or entropy calculations to create preference pairs much more cheaply. Tests on translation and summarization tasks using six different models show SVSI consistently improves over the base model and generally performs as well as or better than self-judging or entropy-based methods. A control experiment confirms the signal is valid, as it degrades performance. The method is also tested as a dense reward for GRPO, where it's competitive but shows higher variance. Ablations show it's robust to different embedders and clustering tools.

**Strengths:**

* The mechanism is simple, general, and easy to explain, acting as a "soft majority vote" without needing a separate judge model.

* Building preference pairs is computationally cheap, and the cost doesn't scale with the size of the base model, unlike LLM-as-a-judge.

* The paper provides broad empirical evidence across multiple models, tasks, and metrics, including a useful "flipped-signal" control test.

* Improvements generalize to out-of-distribution datasets, suggesting the model is learning a real capability, not just overfitting to the test set.

**Weaknesses:**

* The core assumption (consensus = correctness) can be a flaw. The method rewards conformity and could reinforce common errors while discarding correct but niche answers found in minority clusters.

* Some baselines, like self-judging, perform poorly or collapse, making it unclear how much of SVSI's lead is due to its own merit versus the fragility of the alternatives.

* The evaluation is limited to narrow tasks (translation to English, summarization). It hasn't been tested on long-form generation, conversation, safety, or creative tasks. The embeddings used are also English-centric.

* The compute savings focus only on building the pairs, not on the sampling of answers, which is the most expensive part. True end-to-end time savings aren't clear.

* When used as a dense reward for RL, the method seems less stable than discrete rewards.

**Questions:**

* How well does the "top vote" answer actually align with human judgment? A correlation study can be helpful to separate consensus from true quality.

* Could the "majority-error" problem be fixed by retaining multiple clusters? Could "hard negatives" from outside the main cluster be used to improve DPO training?

* How does this method scale to much larger models and longer, more complex outputs? How does it perform on non-English tasks?

---

> ### Author Response · Authors · 2025-11-20
> **Response to Reviewer nFpD (1)**
>
> Thanks for your constructive feedback. Here are some responses with clarification and some new results.
>
> 1. About more tasks. We choose translation and summarization primarily for their well-established metrics (e.g. BLEU, Rouge-L), which provides a relatively objective and fair measurement. We acknowledge that experiments on more tasks would be beneficial. Therefore, we test SVSI on instructions from alpaca-eval, a dataset for general instruction following. We compare the responses before and after self-improvement, and use DeepSeek-V3 as the judge. The Followings are the results. As it shows, SVSI can still achieve general improvement over most of the models.
> | SVSI | Llama-1B | Llama-3B | Llama-8B | Qwen-1.5B | Qwen-3B | Qwen-7B | Avg.   |
> |------|----------|----------|----------|-----------|---------|---------|--------|
> | Win  | 41.25%   | 37.77%   | 39.38%   | 33.66%    | 37.02%  | 40.13%  | 38.20% |
> | Tie  | 22.23%   | 25.34%   | 19.01%   | 34.41%    | 29.81%  | 20.99%  | 25.30% |
> | Lose | 36.52%   | 36.89%   | 41.61%   | 31.93%    | 33.17%  | 38.88%  | 36.50% |
>
> 2. About non-English tasks. We conduct inverse translation (English to other languages) tests over the four translation tasks, with a multilingual embedding model bge-m3 (https://huggingface.co/BAAI/bge-m3). As the results show, SVSI can still achieve self-improvement on back translation tasks.
> | methods | wmt24pp_de_inv | wmt24pp_fr_inv | wmt24pp_ru_inv | wmt24pp_es_inv |
> |---------------|-----------------------|-----------------------|-----------------------|----------------------|
> | Qwen-7B-base  | 12.18                 | 17.44                 | 6.51                  | 20.39                |
> | Qwen-7B-SVSI  | 15.22                 | 19.19                 | 8.75                  | 22.98                |
> | Llama-8B-base | 16.09                 | 23.34                 | 9.90                  | 26.29                |
> | Llama-8B-SVSI | 17.51                 | 23.97                 | 10.57                 | 27.16                |
>
> 3. About computation overhead. We only compare the time cost used on building pairs (scoring) because it is the only time cost difference between SVSI and the two self-evaluation methods (i.e., the baseline methods also need sampling answers and training). For more detailed information, here is an end-to-end computation overhead on Qwen models (Notably, the training time may be much longer than normal because our computational resources are so limited that we have to use cpu-offload). As shown, when taking the sampling time into consideration, the time saved by semantic voting is basically beyond 40\%. For the self-evaluation baseline methods, scoring is not significantly cheaper than sampling since they need additional forward processing on all self-generation.
> |size| Sampling | Scoring(SV/EM/SJ) | Training |
> |------|----------|-------------------|----------|
> | 1.5B | 291      | 40/271/225        | 2723     |
> | 3B   | 575      | 44/479/405        | 4044     |
> | 7B   | 893      | 40/693/669        | 6813     |
>
> 4. About semantic voting on GRPO. As we discussed in section 4.5, although with fluctuation, SVSI-G consistently surpasses two baseline methods. We believe this fluctuation is not a very severe problem, since in most cases even the bottomline of SVSI-G is beyond the topline of baseline. For solutions, an intuitive way is to make the voting score coarse-grained. We discrete the voting score to 1-5 (mapping the top 20\% to 5, the second top 20\% to 4, and so on) instead of using the original numbers. Following are the results of Qwen-1.5B. As we can see, the variance clearly decreases with discrete rewards.
> | methods | wmt24pp_de     | wmt24pp_fr     | wmt24pp_ru     | wmt24pp_es     |
> |-----------------|----------------|----------------|----------------|----------------|
> | SVSI-G-dense    | 17.43$\pm$0.74 | 15.23$\pm$0.48 | 11.33$\pm$0.33 | 21.63$\pm$0.85 |
> | SVSI-G-discrete | 17.63$\pm$0.17 | 15.11$\pm$0.13 | 11.24$\pm$0.03 | 22.30$\pm$0.17 |

---

> > ### Author Response · Authors · 2025-11-20
> > **Response to Reviewer nFpD (2)**
> >
> > 5. About human judgment. We do not include human evaluation mainly because of its subjectivity and irreproducibility. While we do acknowledge that human evaluation provides valuable insights on unverifiable tasks, in practice, it is nearly impossible for us to organize professionals for a fair and comprehensive evaluation within such a short term. We'd like to provide an alternative assessment for your information. MetricX[1] is a well-established reward model evaluating multilingual translation quality, which embodies high consistency with human ratings as reported. We compute Kendall's tau coefficient between semantic-voting-based ranking with that ranked by MetricX, as an indirect reference to consistency with humans. Following are the results. As it shows, semantic-voting-based ranking has a much more significant correlation compared to a random ranking.
> > | rankings | wmt24pp_de | wmt24pp_fr | wmt24pp_ru | wmt24pp_es |
> > |---------------------------|------------|------------|------------|------------|
> > | SV-ranking (Qwen-7B)      | 18.22      | 21.28      | 10.24      | 18.02      |
> > | random-ranking (Qwen-7B)  | 1.55       | 1.25       | 0.46       | 0.56       |
> > | SV-ranking (Llama-8B)     | 23.42      | 23.88      | 17.69      | 24.02      |
> > | random-ranking (Llama-8B) | 0.32       | 0.87       | 0.63       | 0.78       |
> >
> > 6. About retaining multiple clusters. As we discussed in Section 4.6, removing clustering (which equals retaining all clusters) could even slightly improve the performance on Qwen models, but cause a significant drop on Llama models. Therefore, we can infer that retaining multiple clusters may be beneficial when the base model has a relatively strong capability, but not a good idea for less-capable models.
> >
> > 7. About "hard negatives". As we discussed in Section 4.6, it is hard to find meaningful comparison information among different clusters. A feasible implementation of "hard negatives" is those responses with the wrong format. We utilize these "hard negatives" as the dispreferred example in DPO and keep others unchanged. We conduct experiments on Qwen-1.5B and Llama-1B. Following are the results. SVSI-hn means using hard negatives. For simplicity, only BLEU is reported as n-MQM trends are identical to it. Interestingly, using hard negatives is better than original SVSI in Llama-1B, while consistently inferior in Qwen-1.5B. It is possible because using hard negatives will make the model more focused on format compliance rather than semantic difference, which may be helpful when the model often fails to follow the required format. But for a more capable model where adhering to the format is not a problem, this superficial focus does more harm than good.
> > | methods | wmt24pp_de | wmt24pp_fr | wmt24pp_ru | wmt24pp_es |
> > |---------------------|------------|------------|------------|------------|
> > | SVSI (Qwen-1.5B)    | 18.04      | 20.34      | 11.93      | 23.46      |
> > | SVSI-hn (Qwen-1.5B) | 14.07      | 17.44      | 11.77      | 19.62      |
> > | SVSI (Llama-1B)     | 5.28       | 5.59       | 2.75       | 6.89       |
> > | SVSI-hn (Llama-1B)  | 7.02       | 6.31       | 4.24       | 10.04      |
> >
> > Thanks again for your suggestions and we hope our responses will be helpful.
> > ### Reference
> > [1] Juraska, Juraj, et al. "MetricX-23: The Google submission to the WMT 2023 metrics shared task." Proceedings of the Eighth Conference on Machine Translation. 2023

---

### Meta-Review · Area_Chair_WR8t · 2026-01-07

**Summary:**

Reviewers’ major concerns include (1) the core assumption that semantic consensus correlates with correctness, which may introduce conformity bias and suppress diverse but high-quality minority answers; (2) limited evaluation beyond translation and summarization, with questions about generalization to broader open-ended tasks and to non-English settings; (3) incomplete end-to-end efficiency evaluation; (4) sensitivity to the embedding model and clustering choices/hyperparameters; 4) poor writing, and (5) instability when the semantic voting score is used as a dense reward in RL-style training.

**Reviewer Concerns:**

***Addressed by the rebuttal***

Broader tasks: Added alpaca-eval results to test beyond translation andsummarization.

Non-English: Added inverse translation experiments and results using a multilingual embedding model (BGE-m3).

End-to-end compute reporting: Provided end-to-end timing breakdowns and clarified where the savings arise.

Embedding/model choice sensitivity: Added comparisons with multilingual embeddings and cross-encoders; added further ablations.

GRPO instability: Proposed and evaluated a discretized reward variant.

About human evaluation: Added Kendall’s tau correlation (semantic-voting ranking vs. an external translation quality model) as indirect evidence of alignment.

***Still outstanding / partially addressed***

Conformity bias risk: The rebuttal discusses retaining multiple clusters and “hard negatives,” but the fundamental limitation remains, and there is no direct human study on open-ended tasks beyond translation-quality proxies.

Generalization to long-form generation: Cross-encoder results are added, but a comprehensive evaluation on long outputs, dialogue, or more subjective open-ended tasks remains limited.

Novelty: The rebuttal clarifies positioning and value, but reviewers who prioritize conceptual novelty may remain unconvinced.

**Reviewer Scores:**

Reviewer nFpD (initial: 6):
Likely remains positive. The rebuttal directly addressed most questions (more tasks, non-English, compute, correlation, stability) and added substantial new evidence.

Reviewer ikCz (initial: 2 → stated update to 6):
Already raised to 6 in discussion.

Reviewer F69H (initial: 6):
Likely remains at 6. The rebuttal included embedding/cross-encoder results, multilingual tests, additional tasks, and a human-alignment proxy, which addressed most concerns.

Reviewer jSCx (initial: 4 → stated update to 6):
Already raised to 6 after the rebuttal responses on diversity analysis, GRPO stabilization, and discussion of long-text options.

With all reviewers converging to positive scores after the rebuttal and discussion, the AC sees no reason to overturn the reviewers’ evaluations and therefore recommends acceptance.

---

### Decision · Program_Chairs · 2026-01-26

Accept (Poster)